# Endothelial-secreted Endocan activates PDGFRA and regulates vascularity and spatial phenotype in glioblastoma

Soniya Bastola [1,18], Marat S. Pavlyukov[1,2,18], Neel Sharma[1], Yasmin Ghochani[1], Mayu A. Nakano[3], Sree Deepthi Muthukrishnan[1], Sang Yul Yu[1], Min Soo Kim [1,4], Alireza Sohrabi [5], Natalia P. Biscola[6], Daisuke Yamashita[7], Ksenia S. Anufrieva[8,9], Tatyana F. Kovalenko[2], Grace Jung[10], Tomas Ganz [10], Beatrice O'Brien[1], Riki Kawaguchi[1,11], Yue Qin[11], Stephanie K. Seidlits[5], Alma L. Burlingame [12], Juan A. Oses-Prieto [12], Leif A. Havton[13], Steven A. Goldman [14,15], Anita B. Hjelmeland [16], Ichiro Nakano[17,19] ✉ & Harley I. Kornblum [1,19] ✉

Extensive neovascularization is a hallmark of glioblastoma (GBM). In addition to supplying oxygen and nutrients, vascular endothelial cells provide trophic support to GBM cells via paracrine signaling. Here we report that Endocan (*ESM1*), an endothelial-secreted proteoglycan, confers enhanced proliferative, migratory, and angiogenic properties to GBM cells and regulates their spatial identity. Mechanistically, Endocan exerts at least part of its functions via direct binding and activation of the PDGFRA receptor. Subsequent downstream signaling enhances chromatin accessibility of the Myc promoter and upregulates Myc expression inducing stable phenotypic changes in GBM cells. Furthermore, Endocan confers radioprotection on GBM cells in vitro and in vivo. Inhibition of Endocan-PDGFRA signaling with ponatinib increases survival in the *Esm1* wild-type but not in the *Esm1* knock-out mouse GBM model. Our findings identify Endocan and its downstream signaling axis as a potential target to subdue GBM recurrence and highlight the importance of vascular-tumor interactions for GBM development.

Glioblastoma (GBM) is a highly aggressive, vascular-rich tumor that exploits vascular endothelial (VE) cells to contribute to its growth[1,2]. VE cells provide paracrine factors, nutrition, oxygen, as well as a niche for glioma stem cells (GSCs)[1,3,4]. GSCs residing within the perivascular niche maintain the capacity for self-renewal and promote GBM progression, and recurrence[3,5]. Developing an understanding of the basic biology of tumor-associated vasculature will open the door to new therapeutic targets and the establishment of novel therapies.

In experimental models, tumor-associated VE cells support the maintenance of GSCs[3]. This function can be recapitulated by VE cell-derived conditioned media (CM), suggesting that VE cells produce soluble factors that influence the GSC phenotype. In turn, GSCs can

secrete angiogenic factors to promote further neovacularisation[6]. Our recent work has established an interactome network within GBM-associated VE cells and identified key dysregulated genes encoding secreted proteins emanating from VE cells[7]. One such gene is *ESM1* that encodes Endocan, a soluble proteoglycan of 50 kDa, constituted of a mature polypeptide of 165 amino acids (18 kDa) and a single dermatan sulphate chain covalently linked to the serine residue at position 137[8]. *ESM1* expression is regulated by the VEGF/hypoxia pathway and serves as a marker of endothelial cell activation during neoangiogenesis in renal, non-small cell lung, hepatocellular, bladder carcinomas, and GBM[8–10]. *ESM1* is significantly upregulated in tumor-derived vascular cells compared to non-transformed VE cells and its level correlates

with higher grade and shorter survival in glioma[11,12]. However, the functional role of Endocan in glioblastoma and, subsequently, its potential as a therapeutic target remains unclear.

In this study, using patient-derived GBM and VE cells, syngeneic orthotopic GBM models, and an *Esm1* knockout mouse model[13], we investigate the role of Endocan in intercellular crosstalk between GBM and VE cells. Our findings suggest that targeting Endocan itself or downstream signaling pathways activated by this protein may represent a promising therapeutic approach for GBM.

## Results

### Vascular-secreted Endocan promotes proliferation and migration of GBM cells

To understand how VE cells contribute to the progression of GBM, we tested whether VE cells promote the aggressiveness of GBM in vitro and in vivo by using two VE cell models: a short-term primary culture of glioblastoma patient-derived VE cells (TEC15) and an immortalized brain VE cell line HBEC-5i. First, we demonstrated that conditioned media (CM) collected from both TEC15 and HBEC-5i cells promote in vitro proliferation of gliomasphere lines obtained from four different patients (Fig. 1a and Fig. S1a). Next, we co-injected VE cells (TEC15 or HBEC-5i) with the patient-derived IDH wild type gliomasphere line 1051[14] at a 1:10 ratio into SCID mice, and showed that they significantly shortened animal survival (Figs. 1b, S1b). To identify potential trophic factors that could be elaborated by tumor-associated VE cells, we previously isolated CD31+ tumor-associated vascular cells from 8 glioblastoma cases along with non-cancerous CD31+ cells from 5 craniotomies for epilepsy as controls[7]. Subsequent RNA-sequencing (RNA-seq) revealed *ESM1* as the most significantly upregulated gene in tumor-derived CD31+ cells compared to the non-neoplastic counterparts (Fig. 1c). In the TCGA database, *ESM1* expression was found to correlate with higher glioma grade (Fig. S1c) and shorter survival of GBM patients (Fig. S1d).

ELISA analysis demonstrated that Endocan, the *ESM1* gene product, was secreted by HBEC-5i and glioblastoma patient-derived VE cells (TEC 14 and TEC15) at concentrations exceeding 100 pg/ml, while conditioned media (CM) from normal human astrocytes and gliomasphere lines (1079, 157, 711, and 1051) contained nearly undetectable amounts of Endocan (Fig. 1d). We further confirmed this result by multiple approaches. First, using immunohistochemical staining (IHC) we demonstrated that Endocan is exclusively co-localized with CD31 + VE cells in the perivascular areas of our GBM clinical samples (Fig. 1e). Second, we analyzed a single cell RNAseq dataset of cells isolated from GBM tumors[15] that showed a significant level of ESM1 expression in VE cells but not in other cell types within the neoplasm (Fig. S1e). Third, we analyzed a recently published spatial transcriptomic dataset of GBM tumor tissues[16] and demonstrated that *ESM1* expression is highly colocalized with VWF - a marker of endothelial cells[17] Fig. S1f). Finally, analysis of a previously published dataset of GBM-associated VE cells[18] revealed that *ESM1* is expressed in a specific sub-cluster of endothelial cells that was characterized by gene signatures associated with endothelial tip cell formation, tumor angiogenesis, and vascular basement membrane remodeling (Fig. S1g, h).

Next, we investigated whether Endocan could influence GBM phenotype. Incubation with recombinant Endocan (rEndocan) promoted the proliferation of multiple IDH wild type GBM sphere lines (Fig. 1f). As GBM cells are known to migrate along blood vessels[19,20], we also examined the effects of Endocan on GBM cell migration using a previously described hydrogel-based system[21]. rEndocan-treated spheroids derived from 3 different patients had significantly enhanced motility as revealed by both migration length (Fig. 1g, h) and shape factor analysis (Fig. S1i).

Collectively, these findings suggest that Endocan is primarily secreted by endothelial cells and drives both GBM cell proliferation and migration.

### Endocan is essential for establishing the hypervascular phenotype of GBM

To address the potential role of paracrine Endocan signaling in crafting the GBM phenotype in vivo, we utilized wild-type (WT) and *Esm1* knockout (*Esm1* KO) mice[13] as host systems for implanting glioma cells derived from two different murine GBM models. First, we used in vitro gliomasphere cultures termed as 7080[22] that was derived from mice harboring mutations in *p53*, *Pten*, and *Nf1* and enriched for the tumor initiating subpopulation. Second, we utilized freshly-resected murine glioblastoma-like cells isolated from tumors induced by RCAS-PDGAB lentivirus injection into Nestin-Tva/Cdkn2a-/- mice[23]. Both types of cancer cells were injected into the brains of WT and *Esm1* KO mice. The resultant tumors in WT mice exhibited numerous intratumoral hemorrhages in sharp contrast to grayish necrotic tumors in *Esm1* KO mice (Figs. 2a, S2a). Subsequent IHC for CD31 demonstrated significantly fewer CD31+ VE cells in the tumors from *Esm1* KO animals with reduced vessel diameter as compared to those formed in wildtype animals (Figs. 2a, b, S2a, b). At the ultrastructural level, 7080 tumors in the WT host exhibited proliferative blood vessels with hypertrophy and the basement membrane showed an irregular outline and varying diameters, typical of human glioblastoma[19,24], while in the *Esm1* KO host, tumors exhibited sparse blood vessels with a regular basement membrane measured as Blood vessel ratio-SAE using previously defined methods[25] (Fig. 2c, d). Consistent with these observations, qRT-PCR analysis demonstrated that vascular cell-associated genes known to play an integral part in maintaining barrier function, adhesion interactions, and tight junctions(*Podxl*, *Cld5*, and *Cdh5*)[26,27], along with a marker for endothelial cells (*Pecam1*)[18] were downregulated in tumors from *Esm1* KO mice (Figure S2d). Importantly, brain vascularization in the areas that did not contain tumor was not appreciably different between wild type and *Esm1* knockout mice, as can be seen from CD31 staining (Fig. S2c) and electron microscopy (Fig. 2c, d). Next, we compared immune cells infiltration in tumors formed in *Esm1* KO and WT animals. IHC staining for microglia and macrophage (Iba1,Cd68 and F4/80)[28] and T-cell (Cd8)[29] markers showed no statistically significant differences between Esm1 WT and KO tumors Fig. S2e.

We then performed RNA-seq expression profiling of tumors formed by 7080 cells in WT and *Esm1* KO mice and identified 1449 downregulated and 283 upregulated genes (fold change >4; adjusted *p* < 0.05) in KO samples. Subsequent Gene Set Enrichment analysis (GSEA) revealed alterations in many important pathways including downregulation of proliferation, DNA repair as well as Myc and VEGFA signaling in tumors formed in *Esm1* KO mice (Fig. S3a and Supplementary Data Table 1).

Thus far, our findings indicate that Endocan promotes the hypervascularization phenotype of murine gliomas. This effect can be mediated either by autocrine (GBM cells promote Endocan expression in VE cells, which further enhance proliferation of VE cells) or paracrine (GBM cells treated with Endocan secrete other factors which promote vascular cell growth) mechanisms, or a combination of the two. In order to discriminate between these hypotheses, we first treated two human TEC cell lines with rEndocan and in both cases observed a significant increase in proliferation in response to Endocan (Fig. 2e). Next, to test the potential for an indirect effect of Endocan on endothelial cells, we treated GBM cells with rEndocan for 3 days, and then replaced the rEndocan containing media with regular GBM media. After 4 more days of incubation, conditioned media was collected and added to TECs. Results of this experiment demonstrated significantly higher proliferation of TECs in Endocan-induced conditioned media compared to the control media (Fig. 2f).

Altogether our data suggest that Endocan is essential for the development of neovascularization in GBM. This protein, secreted by tumor-associated VE cells, can directly enhance proliferation of VE cells within GBM and also can stimulate GBM cells to secrete other factors additionally promoting vascularization of the malignancy.

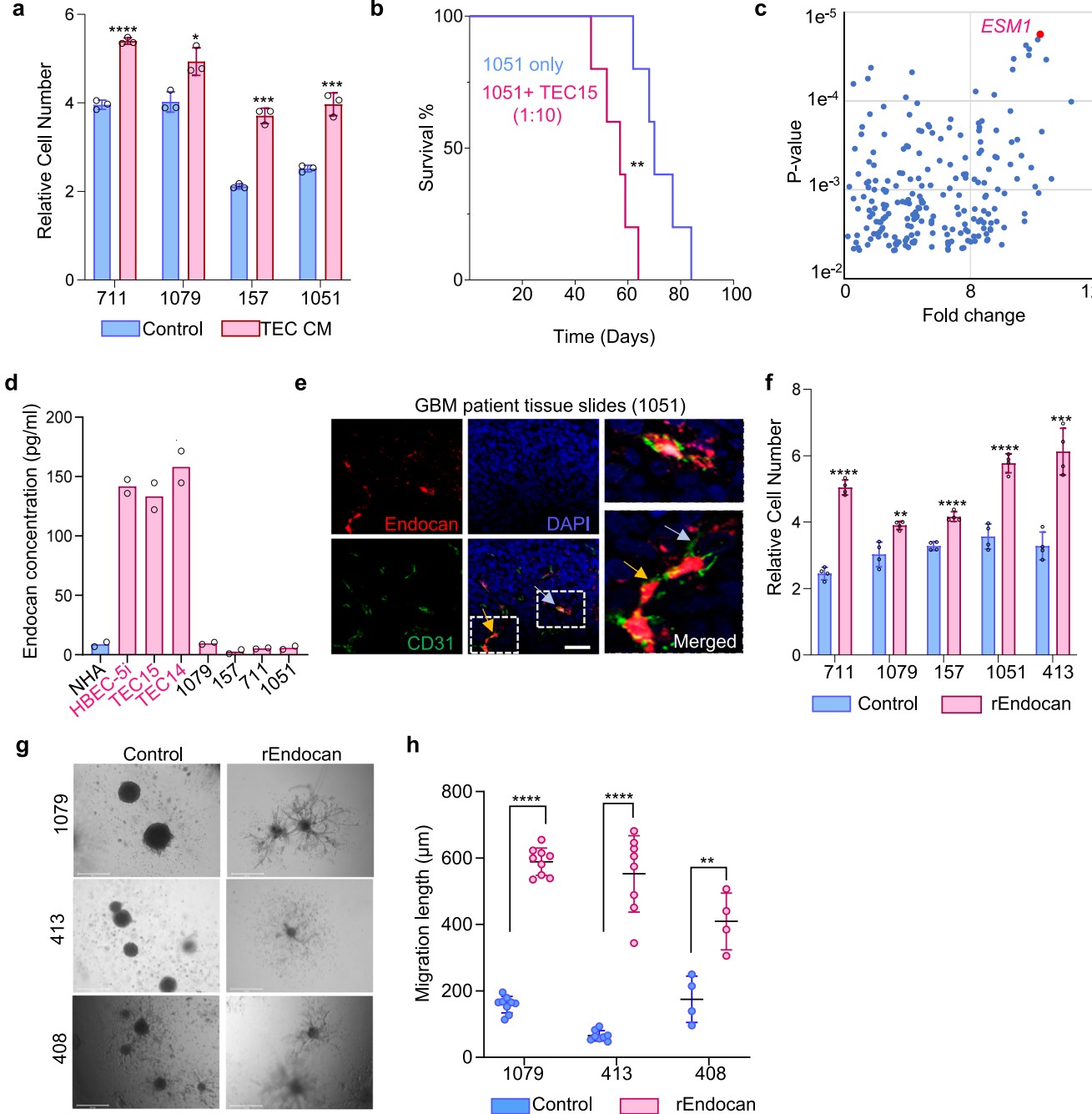

**Fig. 1 | Vascular-secreted Endocan promotes proliferation and migration of GBM cells. a** Proliferation of GBM cells (711, 1079, 157, 1051) treated with control media or conditioned media collected from Tumor Endothelial Cells (TEC CM); $n = 3$ biological replicates, \*\*\*$P < 0.0001$ for 711 cells, \*$P = 0.015$ for 1079 cells, \*\*\*$P = 0.0001$ for 157 cells, \*\*$P = 0.0007$ for 1051 cells, unpaired two-tailed Student's *t*-est. **b** Kaplan–Meier survival analysis of SCID mice intracranially injected with gliomaspheres alone (1051) or gliomaspheres with VE cells (1051 + TEC) at a 10:1 ratio. ($n = 5$ mice per group). \*\*$P = 0.0064$, Log-rank (Mantel-Cox) test. **c** Volcano plot of RNA-sequencing (RNA-seq) data comparing gene expressions of normal and tumor endothelial cells, with ESM1 (red font). FC = 9.76, FDR corrected $p = 0.0031$. Original data was used from previously published study[7]. **d** ELISA assay comparing levels of secreted Endocan in CM from Normal Human Astrocytes (NHA), Human Brain Endothelial Cells (HBEC-5i), Tumor-associated Endothelial cells (TEC15, TEC14), and GBM cells (1079, 157, 711, and 1051), $n = 2$ biological replicates. **e** Representative immunofluorescence (IF) staining for Endocan (red) and CD31 (green) in formalin-fixed paraffin-embedded tissue sections from glioblastoma patient 1051. Nuclei stained with DAPI (blue). Scale bar, 50 μm. $n = 3$ different patients. **f** Proliferation of GBM cells (711, 1079, 157, 1051, 413) with or without rEndocan (10 ng/ml). Proliferation was measured 5 days after treatment; $n = 4$ biological replicates; \*\*\*\*$P < 0.0001$ for 711 cells, \*\*$P = 0.0043$ for 1079 cells, \*\*\*\*$P < 0.0001$ for 157 cells, \*\*\*\*$P < 0.0001$ for 1051 cells, unpaired two-tailed Student's *t* test. **g** Representative microscopic images of control and rEndocan treated (10 ng/ml) GBM spheroids encapsulated in the hydrogel. Images were taken 3 days post-encapsulation. Scale bar, 500 μm. **h** Quantification of migration distance in control and rEndocan (10 ng/ml) spheroids formed by 408, 413, or 1079 cells. $n = 9$ spheroids for 408 cells, $n = 8$ spheroids for 413 cells, and 4 spheroids for 1079 cells were used for analysis \*\*\*\*$P < 0.0001$ for 408 cells, \*\*$P = 0.0053$ for 413 cells and \*\*\*\*$P < 0.0001$ for 1079 cells, unpaired two-tailed Student's *t* test. Source data are available in the Source Data file.

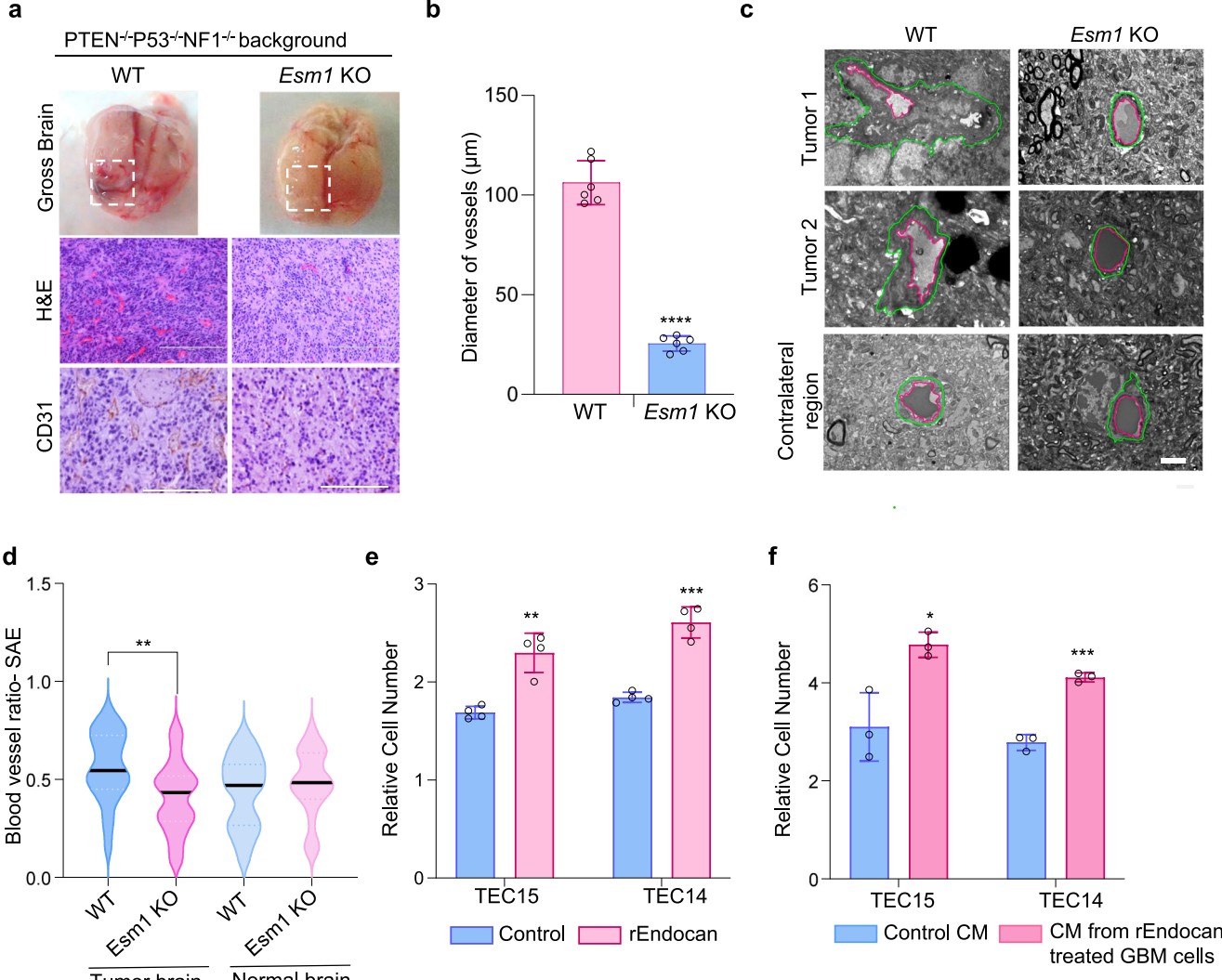

**Fig. 2 | Endocan is essential for establishing the hypervascular phenotype of GBM. a** Images of the brains collected from 7080 glioma-bearing *Esm*1 WT or *Esm1* KO mice (upper panel). H&E staining of the tumor slices (middle panel, Scale bar, 200 μm) and CD31 staining of the tumor slices (lower panel, Scale bar, 200 μm). **b** Quantitation of the diameter of the blood vessels formed in tumors from *PTEN/P53/NF1* deleted glioblastoma model. CD31 positive VE cells were imaged and then measured using ImageJ vessel analysis tool. $n = 7$ regions from $n = 5$ mice were used for the analysis, $****P < 0.0001$, unpaired two-tailed Student's *t* test. **c** Representative transmission electron microscopy images of tumors formed in *Esm1* WT and KO mice (middle panel) and contralateral tissue region from non-tumor hemisphere (lower panel). Green and magenta lines indicate the outer and inner diameter of the blood vessels respectively. Scale bar, 2 μm. **d** Violin plots show frequency of blood vessel ratio of Shape Adjusted Eclipse (SAE) distributions (y-axis). Center lines denote median values obtained by the SAE approach after determining vessel area or perimeter of blood vessels in the tumor-bearing hemisphere of mice brains and contralateral region. $**P = 0.0030$, One-way ANOVA followed by post hoc test. **e** Proliferation of rEndocan treated (10 ng/ml) and untreated TEC cells (TEC 15 and TEC 14). Proliferation was measured on Day 5 post-treatment, $n = 4$ biological replicates, $**P = 0.0012$ for TEC15 and $***P = 0.0013$ for TEC14, unpaired two-tailed Student's *t* test. **f** Proliferation of TEC15 and TEC14 cells treated with conditioned media collected from the control GBM cells or GBM cells pretreated with rEndocan (10 ng/ml) for 3 days. Proliferation was measured on day 5 post-treatment, $n = 3$ biological replicates, $*P = 0.0173$ for TEC15 and $***P = 0.0002$ for TEC15, unpaired two-tailed Student's *t* test. Source data are available in the Source Data file.

## Endocan induces stable phenotypic alterations of GBM cells regulating their spatial identity

Next, we aimed to assess if GBM cells propagated in vivo in a micro-environment with or without Endocan acquire any stable differences in their phenotype. To answer this question, we generated sphere cultures from tumors formed in WT and *Esm1* KO mice brains (WTD and KOD cells respectively). Even under identical in vitro cultivation conditions, KOD GBM cells had reduced proliferation compared to WTD counterparts. Furthermore, KOD cells no longer responded to the addition of rEndocan (Fig. 3a). To further characterize differences between these cells, we labeled WTD and KOD cells with GFP and mCherry, respectively, and co-injected them into the brains of *Esm1* WT mice (Fig. 3b). IHC analysis of the resultant tumors demonstrated

that WTD cells were located at the tumor edge region and infiltrated into the normal brain, whereas KOD cells appeared to be largely confined to the tumor core (Fig. 3c, d). To quantitatively assess this phenomenon, we measured the number of cells of each type as a function of the distance from the injection site, demonstrating that KOD cells were mainly present in the tumor core-region, whereas WTD cells were found to be highly migratory and spread across the tumor and normal brain (Fig. 3e). Interestingly, CD31 staining indicated that the core lesion predominantly formed by KOD cells was hypovascular, while the edge lesion created mainly by WTD cells was hypervascular (Fig. 3f). The tumor core areas exhibited higher immunoreactivity to hypoxic and mesenchymal GBM markers (HIF1α, YKL-40, p-65 NF-κB and ALDH1A3)[30–33] (Figs. 3f and S3b), while the edge regions had higher

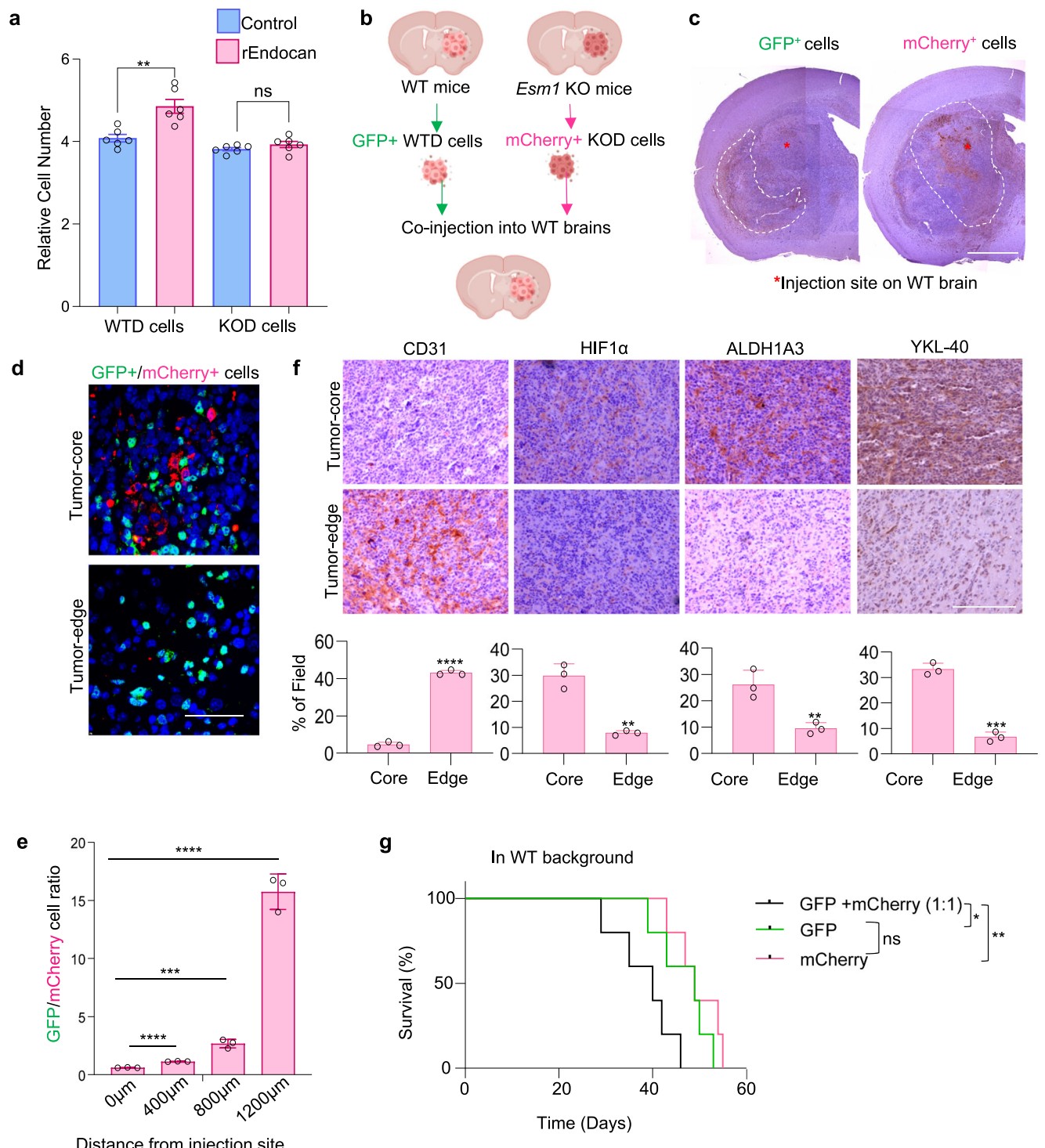

levels of proneural markers PDGFRA and Olig2 (Fig. S3b). Taken together, these findings suggest that KOD cells have lost their capacity to promote neovascularization and to invade into the normal tissues even in the presence of Endocan.

Surprisingly, we did not observe any survival differences between mice injected with WTD or KOD cells (Fig. 3g). However, when a mixture of KOD and WTD cells was co-injected into the brain, animal survival was substantially reduced compared to the mice injected with either KOD or WTD cells alone. One possible explanation for this result could be that mixing of KOD and WTD cells together enhances cancer cell heterogeneity within the developing tumors, which has been

shown to promote GBM malignancy by various molecular mechanisms[27,34]. However, additional experiments will be needed to test this hypothesis.

Altogether, the striking differences observed between WTD and KOD cells suggest that Endocan itself or the tumor microenvironment formed in the presence or absence of Endocan has a significant and long-lasting effect on the key vascular and infiltrative properties of GBM cells. Importantly, the phenotypical differences developed between cells propagated with or without Endocan can be stably maintained even if the cells are subsequently cultured under identical conditions in vitro or in vivo.

**Fig. 3 | Endocan induces stable phenotypic alterations of GBM cells regulating their spatial identity. a** Proliferation assay of control and rEndocan treated (10 ng/ml) WTD and KOD cells. **$P = 0.0023$ for WTD cells and $P = 0.2104$ for KOD cells, $n = 6$ biological replicates, unpaired two-tailed Student's $t$ test. **b** Schematic figure delineating the steps for the experiment. Briefly, mouse 7080 spheres were implanted into brains of WT and *Esm1* KO host, once tumors were formed, they were isolated into single cells, and were followed by labeling *Esm1* KO derived spheres with mCherry and WT derived spheres with GFP. Cells were then co-injected into WT host ($n = 5$ mice per group). The figure was created in BioRender. Kornblum, H. (2024) BioRender.com/n19r944. **c** Representative IHC staining with anti-GFP (left) and anti-mCherry (right) antibodies of mice brains co-injected with WTD (GFP$^+$) and KOD (mCherry$^+$) cells ($n = 5$ mice per group). The boundaries of the staining areas are shown by white lines, injection site indicated by red asterisk. Scale bar, 2 mm. **d** Representative IHF images of mice brain sections stained with anti-GFP (green), anti-mCherry (red) antibodies and DAPI (blue). $n = 3$ biological replicates, Scale bar, 50 μm. **e** Ratio of GFP+ and mCherry+ cells in IHF images as in "D" at different distances from the site of injection. Quantification was performed using ImageJ software where mCherry+ cells and GFP+ cells were divided by DAPI staining. The experiment was performed using $n = 3$ mice per group. ****$P < 0.0001$ for 400 μm vs. 0 μm, ***$P = 0.0006$ for 800 μm vs. 0 μm, and ****$P < 0.0001$ for 1200 μm vs. 0 μm, unpaired two-tailed Student's $t$ test. **f** Representative IHC images of core (upper panel) and edge regions (middle panel) of the tumor obtained as in "D" stained for CD31, HIF1α, ALDH1A3, and YKL-40. % of field positive areas for indicated proteins on lower panel ($n = 3$ mice). **$P < 0.0001$ for CD31, **$P = 0.0013$ for HIF1α, ***$P = 0.0083$ for ALDH1A3 and ****$P = 0.0001$ for YKL-40, unpaired two-tailed Student's $t$ test. *Scale* bar, 200 μm. **g** Kaplan–Meier survival analysis of *Esm1* WT mice intracranially injected with WTD and KOD cells or the mixture of both at 1:1 ratio. ($n = 5$ mice per group), *$P = 0.047$; ns. - nonsignificant ($P = 0.31$); **$P = 0.00642$. P values were determined by log-rank test. Source data are available in the Source Data file.

## Endocan binds to and activates PDGFRA in human glioblastoma cells

As a next step, we sought to understand the relevant mode of action of Endocan in glioblastoma. To identify possible receptors for Endocan on GBM cells, we purified integral membrane proteins from human gliomasphere line 157 and incubated these proteins with immobilized rEndocan. Subsequent elution of bound proteins and mass spectrometry identified PDGFRA as a potential Endocan-interacting protein (Supplementary Data Table 2). To confirm this finding in a cell-free system, we tested the binding of the corresponding recombinant proteins. Pull down experiments demonstrated physical interaction between F$_c$-tagged PDGFRA monomeric fragment and His-tagged rEndocan (Fig. 4a). Next, we used surface plasmon resonance (SPR) to evaluate the binding affinity of recombinant His$_6$-Endocan to a recombinant Fc-PDGFRA monomeric fragment (Fig. S4a) and found a K$_D$ of $345 \pm 25$ nM. Although this value is orders of magnitude higher than the one that was determined using standard Scatchard assay with radioactively labeled PDGF-BB and cells overexpressing exogenous PDGFRA, it is comparable to the K$_D$ which were previously identified using a similar SPR approach for the interaction of a recombinant PDGFRa with PDGF-AA (600 nM), PDGF-BB (420 nM) and PDGF-AB (840 nM)[36]. As another validation step, we used the proximal ligation assay[35], which showed that rEndocan induces the phosphorylation of endogenous PDGFRA - the key step in PDGFRA activation[36] (Fig. S4b). To further confirm the specificity and affinity of Endocan-interaction with PDGFRA, we performed a binding competition assay between rEndocan labeled with Alexa488 and recombinant PDGF-BB (one of the cognate ligands for PDGFRA), labeled with Alexa647. The addition of rPDGF-BB diminished rEndocan binding to glioblastoma cells in a concentration-dependent manner. Similarly, rEndocan decreased rPDGF-BB binding. Importantly, a concentration of 10 ug/ml rEndocan was able to completely displace rPDGFββ from GBM cells, while rPDGF-BB at the corresponding concentration was able to displace only half of the rEndocan molecules (Fig. 4b), suggesting that Endocan can interact with other receptors in addition to PDGFRA.

Given that PDGFRA activation by functional ligands induces receptor autophosphorylation, we investigated whether the addition of rEndocan alters the phosphorylation levels of endogenous PDGFRA in human gliomaspheres in vitro using PDGF-BB as a positive control[37]. Dose-response analyses demonstrated that both rPDGF-BB and rEndocan induced PDGFRa phosphorylation in brain tumor cells at low nanomolar concentrations. This result is consistent with the K$_D$ value previously determined for PDGF-BB interaction with exogenously expressed full length PDGFRa on the cell surface[39]. Interestingly, the magnitude of the effect of rEndocan was somewhat smaller than that of PDGF-BB (Figs. 4c and S4c). We also observed that in some western blots phosphorylated PDGFRA appeared as multiple bands, and the lower molecular weight bands were especially noticeable when cells were incubated with rPDGF-BB, but not Endocan. These could

potentially be explained by differences in PDGFRA internalization and subsequent degradation induced by rPDGF-BB and Endocan. Next, we performed a time course experiment and found that rEndocan (10 ng/ml, 400 pM) was able to induce appreciable autophosphorylation of PDGFRA in as early as 15 min and displayed similar temporal kinetics to that of rPDGF-BB (Figs. 4d and S4d). rEndocan treatment also resulted in the phosphorylation of the PDGFRA downstream targets such as p85α and PI3K[38] along with phosphorylation of ERK1/2[39] (p44/42 MAPK)[40], further confirming the activation of the PDGFRA signaling pathway (Figs. 4d and S4d). Interestingly, comparing the effects of Endocan and PDGF-BB we noticed substantially higher levels of ERK1/2 phosphorylation upon incubation with PDGF-BB. ERK1/2 activation is an integral part of the canonical PDGFRA RTK signal transduction pathway[41] that has been shown to promote mitotic progression and proliferation[42]. The weaker effect of Endocan on ERK1/2 may be associated with the different kinetics of its interaction with PDGFRA[43]. However, further studies are needed to clarify the significance of the potentially differential activation of the ERK pathway by Endocan and PDGF-BB.

Interaction of PDGFRs with their ligands has been shown to be affected (both enhanced[44] and inhibited[45]) by glycosaminoglycans. To determine if glycosaminoglycans (including dermatan sulfate chain of Endocan) can modulate Endocan binding to PDGFRA we repeated the SPR assay in the presence of heparin and also evaluated the effect of the heparin on Endocan-induced PDGFRA phosphorylation. Both experiments showed no effect of heparin on Endocan-PDGFRA interaction (Fig. S4e, f).

It has been previously reported that Endocan can activate EGFR signaling in non-small cell lung cancer[46]. However, in our experimental conditions we did not detect any Endocan-induced EGFR phosphorylation even at 100 ng/ml concentration of rEndocan (Fig. S5). Interestingly, we observed that in 1079 cells, EGF was able to induce PDGFRa phosphorylation. This result is consistent with previous findings demonstrating that in some patient-derived GBM cell lines PDGFRa and EGFR form heterodimers which can be activated by EGF[47].

Next, we studied how Endocan and PDGF-BB affect GBM cells at the transcriptome level. RNA-seq and subsequent GSEA revealed that their effect on GBM cells was similar but not precisely the same (Fig. 4e), indicating that while Endocan can activate the PDGFRA receptor, it might also have effects that are distinct from those of other PDGFR ligands, possibly due to the binding with other cellular receptors. The pathways most substantially affected by Endocan included angiogenesis, upregulation of Myc targets, and receptor tyrosine kinase (RTK) pathway activation.

To better understand the potential relationships amongst Endocan and other PDGFRa ligands, we analyzed a recently published spatial transcriptomic dataset of GBM tumor tissues[16]. Surprisingly, we found that *ESM1* and *PDGFB* are expressed in the same tumor regions, while the distributions of *PDGFA* and *PDGFC* seem to be mutually

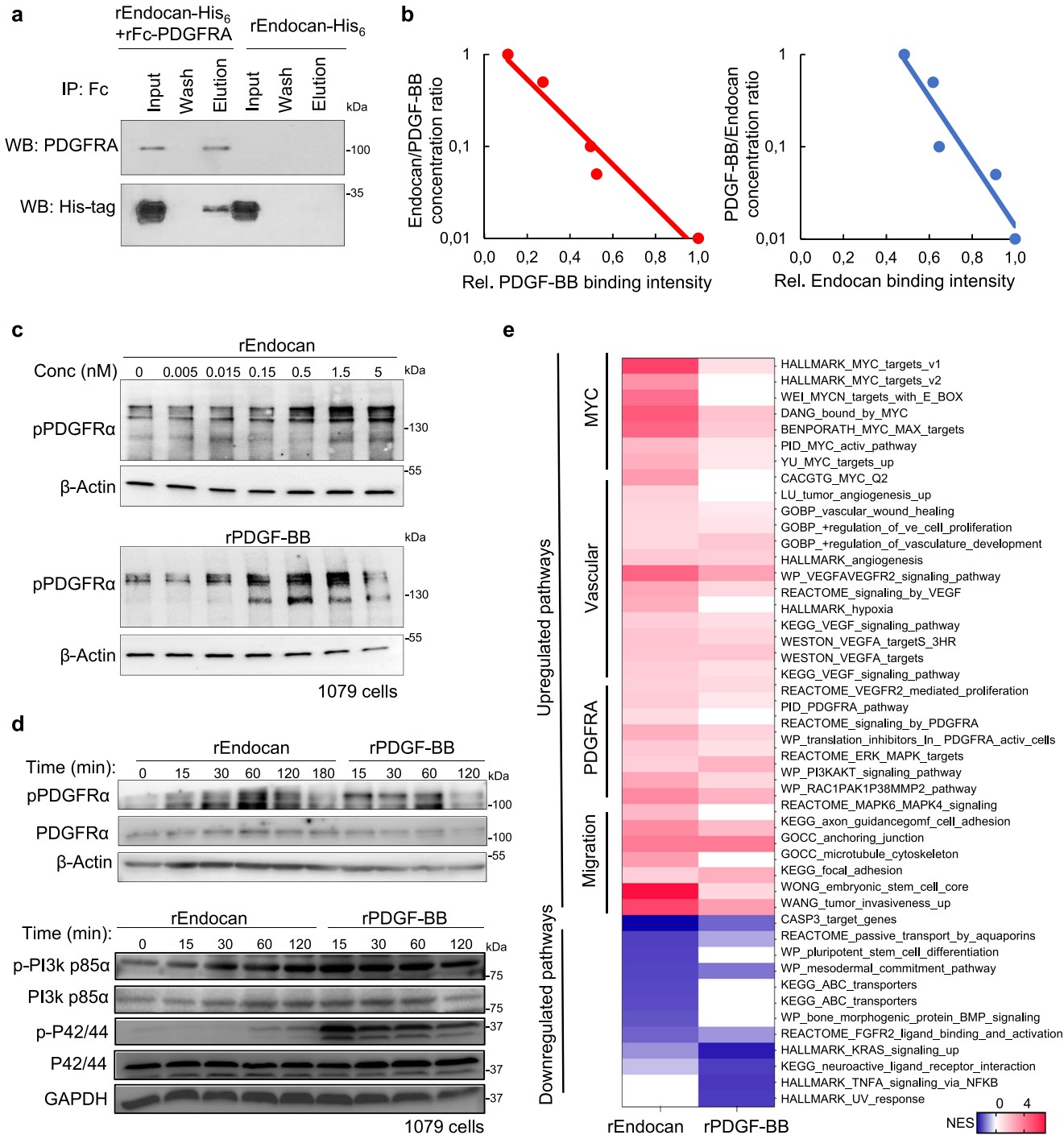

**Fig. 4 | Endocan binds to and activates PDGFRA in human glioblastoma cells.**
**a** Western blot (WB) analysis of samples obtained during pull down assay of His-tagged rEndocan and Fc-tagged rPDGFRA. Representative result of n = 3 biological replicates. **b** Competition binding assay of Endocan-Alexaflour488 and PDGF-BB-Alexaflour647. 1079 cells were incubated with indicated concentrations of both proteins and subsequently analyzed by flow cytometry, n = 3 biological replicates with similar results. **c** Western blot analysis of 1079 cells incubated with different concentrations of rEndocan or rPDGF-BB for 30 min. Representative result of n = 3 biological replicates. **d** Western blot analysis of 1079 cells incubated with 10 ng/ml rEndocan or rPDGF-BB (10 ng/ml) for the indicated period of time. Representative result of n = 3 biological replicates. **e** Heatmap showing significant ($p < 0.05$) normalized enrichment scores (NES) calculated from RNAseq data of 1079 cells treated with 10 ng/ml rEndocan or rPDGF-BB for 48 h, n = 4 biological replicates. Source data are available in the Source Data file.

exclusive with those of PDGFB and ESM1 (Fig. S6a, b). The reason for this is unclear, but we speculate that Endocan may act together with PDGF-BB to maintain a high level of PDGFRa activation in the same tumor region, while other PDGFR ligands are expressed in the different GBM areas and may therefore affect other populations of GBM cells.

Altogether, our data indicate that one of the molecular mechanisms by which Endocan affects GBM is binding to PDGFRA and subsequent activation of the downstream signaling pathways in tumor cells adjacent to VE cells in the microvasculature proliferation area.

## Endocan induces changes in the chromatin structure in the *Myc* promoter region

Since we demonstrated that GBM cells propagated in *Esm1* KO and WT mice showed substantial and highly persistent phenotypic differences,

we hypothesized that Endocan-PDGFRA signaling can induce stable epigenetic alterations. To test this hypothesis, we performed ATAC-seq (Assay for Transposase-Accessible Chromatin sequencing) with *Esm1* WTD and KOD cells[48]. We identified 294 genes that had significantly different chromatin accessibility levels between samples, and among them, 129 genes showed different chromatin structure in their promoter regions (Fig. 5a and Supplementary Data Table 3). The strongest alterations were observed for the *Myc* promoter that showed more than 10-fold increased accessibility in WTD as opposed to KOD cells. We next performed de novo motif analysis of the detected ATAC-seq peaks and identified the motifs for PATZ1 (E-value = $9.6 \times 10^{-83}$) and Irf1 (E-value = $1.2 \times 10^{-80}$) - transcription factors whose expression was shown to be regulated by Myc[49,50] (Fig. S6c).

To explore the relationship between open chromatin structure and gene expression levels, we next performed RNA-seq analysis of the same samples (Fig. 5b, c and Supplementary Data Table 4). Comparison of ATAC-seq and RNA-seq data revealed that among all genes with altered chromatin structure, *Myc* had the most substantial upregulation in expression, with WTD tumors showing 25-folds higher level of *Myc* mRNA than KOD samples. IHC of WTD tumors revealed higher expression of c-Myc as compared to *Esm1* KOD samples (Fig. 5d). To further confirm these data, we performed WB analysis for PDGFRA and Myc in the corresponding cells and saw consistent loss in the expression of both of these proteins in *Esm1* KOD samples (Fig. 5e). To validate our finding that Myc is a downstream target of Endocan-driven PDGFRA signaling, we first treated our WTD and KOD cells with 10 ng/ml of rEndocan for 3 days and found robust upregulation of *Myc* expression in WTD cells but not in *Esm*1 KOD cells (Fig. 5e). Next, we used in vitro culture of patient-derived GBM cells (1079 and 413) and confirmed that Myc was significantly upregulated at both the mRNA and protein levels in response to addition of the rEndocan to all tested cells (Figs. 5f, g, S6d).

### Inhibition of the PDGFRA pathway attenuates the effect of Endocan on GBM cells

We next investigated whether the inhibition PDGFRA signaling would affect Myc expression and functional effects of Endocan. First, using lentiviral knockdown, we demonstrated that Myc level was substantially reduced upon depletion of PDGFRA (Fig. 6a). Knockdown of PDGFRA also abrogated the proproliferative effect of Endocan on GBM cells (Fig. 6b). Second, we utilized small molecule PDGFRA inhibitors ponatinib[51] and nintedanib[52] as well as PDGFRA neutralizing antibodies olaratumab[53]. Our results demonstrated that all three compounds abrogate the effect of Endocan on PDGFRA activation and diminish Myc expression (Figs. 6c and S7a–d). Treatment of GBM cells with ponatinib also reduced the proproliferative effect of rEndocan (Fig. 6d).

Based on these data, we hypothesized that blockade of PDGFRA signaling in the WT background would attenuate Endocan-mediated activation of PDGFRA and subsequent Myc upregulation, leading to a hypovascular phenotype similar to that observed in *Esm1* KO tumors (Fig. 2). To this end, we treated WT and *Esm1* KO mice bearing 7080 tumor cells with ponatinib. Consistent with our hypothesis, ponatinib enhanced survival only of the tumor-bearing WT, but not *Esm1* KO mice (Fig. 6e), and reduced blood vessel formation, as can be seen from both CD31 immunostaining intensity (Fig. 6f) and quantification of the blood vessels diameter (Fig. 6g). Furthermore, IHC staining demonstrated that treatment with ponatinib resulted in a loss of Myc and Olig2 expression accompanied by increased level of p65-NFkB, a marker associated with tumor core (hypovascular) cells[30] (Fig. S7e). Importantly, this effect was only observed in *Esm1* WT but not KO mice.

Taken together, our findings indicate that Endocan-induced PDGFRA activation affects chromatin structure of the Myc promoter region leading to the prolonged upregulation of Myc expression and

stable phenotypic alterations of GBM cells. Importantly, this pathway can be inhibited by small molecule compounds targeting PDGFRA. Tumors formed in the absence of Endocan utilize other mechanisms to drive their growth and are less sensitive to PDGFRA inhibitors.

### Endocan protects glioblastoma cells from radiotherapy

Previous studies have demonstrated that the tumor vascular niche provides protection to GBM cells against radiotherapy (IR)[54]. Therefore, we sought to determine the role of Endocan signaling in this process. First, we demonstrated that IR substantially upregulated *ESM1* transcription in VE cells in vitro as measured by qRT-PCR (Fig. S8a) and increased secretion of Endocan protein by more than 2-fold as determined by ELISA (Fig. 7a). Given these findings, we assessed whether IR-induced upregulation of Endocan could play a role in radioprotection of GBM cells by VE cells. First, we pretreated GBM cells with rEndocan or HBEC-5i conditioned medium (CM) for 3 days, followed by IR at 8 Gy. Analysis of gliomaspheres on day 5 post-irradiation revealed that both rEndocan and HBEC-5i CM significantly decreased IR-induced apoptosis (Figs. 7b and S8b), prevented cell-cycle arrest (Fig, S8c), and promoted GBM cell survival after radiation treatment (Fig. S8d). Furthermore, cells pretreated with Endocan had significantly reduced γ-H2AX staining intensity compared to the control groups (Fig. 7c), indicating diminished IR-induced DNA damage[55].

Given that radiation is known to promote mesenchymal differentiation of gliomaspheres[30], we examined the effect of Endocan on the expression levels of mesenchymal (CD44) and proneuronal (CD133) GBM markers in irradiated 1051 cells. Pretreatment with either HBEC-5i CM or rEndocan suppressed radiation-induced downregulation of CD133 and upregulation of CD44 (Fig. S8e), and therefore, protected GBM cells from mesenchymal differentiation. In our recent study we demonstrated that radiation can also promote acquisition of vascular endothelial-like and pericyte-like phenotypes by GBM cells[53]. However, treatment with rEndocan didn not show any reproducible effect on the levels of endothelial (CD31) and pericyte (Desmin and ACTA2) markers in GBM cells (Fig. S8f).

To confirm that the protective effect of the secretomes from VE cells was specifically mediated by Endocan protein but not by other factors, we depleted Endocan from HBEC-5i CM using an anti-Endocan antibody. This depletion resulted in a decrease in the radioprotection of the HBEC-5i CM as measured by caspase 3/7 activity assay and a cellular proliferation assay (Figs. 7d, S8g). RNA-seq of gliomaspheres that were pretreated with either HBEC-5i CM or rEndocan and subsequently exposed to 8 Gy IR demonstrated that rEndocan largely recapitulated the effects of HBEC-5i CM on the transcriptome of gliomaspheres after irradiation (Fig. 7e). These findings suggest that Endocan is the major mediator for the VE cell-driven radioprotective effect on glioblastoma cells.

Finally, we investigated the effect of radiation treatment on survival of tumor-bearing *Esm1* WT and KO mice. Both tumor models showed substantial shrinkage in size after radiation. However, radiation treatment resulted in a significantly greater enhancement of survival of tumor-bearing KO mice than WT mice (Figs. 7f, S8h). These results are in positive agreement with our RNAseq data that demonstrated downregulation of the DNA repair pathway in tumors formed in *Esm1* KO mice (Fig. S3a).

Altogether, our findings demonstrate that Endocan is a key mediator of the radioprotective effect of VE cells on GBM cells. Importantly, Endocan secretion is additionally increased upon IR which further promotes GBM cell viability and prevents radiation-induced mesenchymal differentiation of glioblastoma cells.

## Discussion

More than 50 years ago, it was demonstrated that malignant tumors promote microvascular proliferation, which in turn provides nutritional support that enables further cancer progression[56]. However,

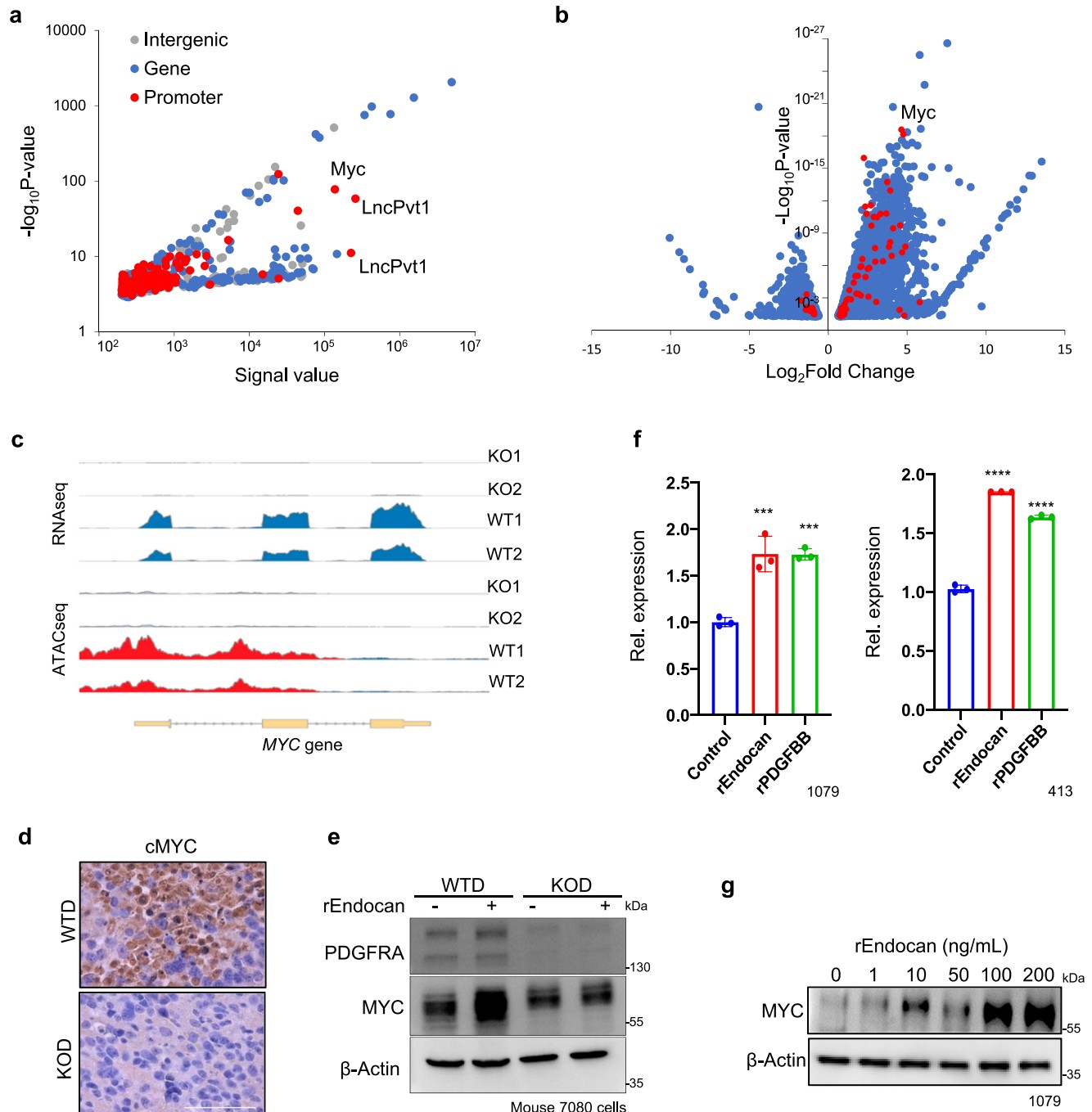

**Fig. 5 | Endocan induces changes in the chromatin structure in the *Myc* promoter region. a** ATAC-seq analysis of tumors derived from *Esm1* WT and KO mice ($n = 2$ mice per group). Individual dots represent peaks located in gene promoter region (red), in transcribed sequence (blue) and outside of the gene (gray). **b** Volcano plot showing differentially expressed genes (LogFC expression ($p < 0.05$) between tumors derived from *Esm1* WT and KO mice as determined by RNAseq analysis. ($n = 3$ mice per group). Genes with altered chromatin structure of their promoter region as determined by ATAC-seq analysis are indicated with red dots. **c** Sequencing read coverage of *Myc* gene as determined by RNA-seq (gene expression; blue) and ATAC-seq (chromatin accessibility; red) of the tumors as in "A" ($n = 2$ mice per group). **d** Representative microscopic images of WTD and KOD

tumor slides stained with anti-MYC antibody. $n = 3$ biological replicates. Scale bar, 100 μm. **e** Western blot analysis of WTD and KOD cells treated or untreated with rEndocan (10 ng/ml) for 3 days. Representative result of $n = 3$ biological replicates. **f** qRT-PCR analysis of *MYC* expression in control and rEndocan (10 ng/ml) or rPDGFBB (10 ng/ml) treated cells human GBM cells (1079 and 413 cell lines). ***$P = 0.0005$ for rEndocan vs. Control and ***$P = 0.0005$ for rPDGFBB vs. Control in 1079 cells. ****$P < 0.0001$ for rEndocan vs. Control and rPDGFBB vs. Control in 413 cells. $n = 3$ biological replicates, One-way ANOVA with Dunnett's multiple comparisons test. **g** Western blot analysis for MYC and β- actin protein in 1079 cells treated with increasing doses of rEndocan for 72 h. Representative result of $n = 3$ biological replicates. Source data are available in the Source Data file.

during the last decade, it has become clear that the relationship between tumor and VE cells is far more complicated. They form a sophisticated multilayered interaction network that shapes the key properties of both cell types. Although many components of this

crosstalk have been identified, the key factors that orchestrate the entirety of tumor-vasculature signaling are not well elucidated.

In the current study, we investigated some of the mechanisms of bidirectional interactions between GBM and VE cells. A dataset

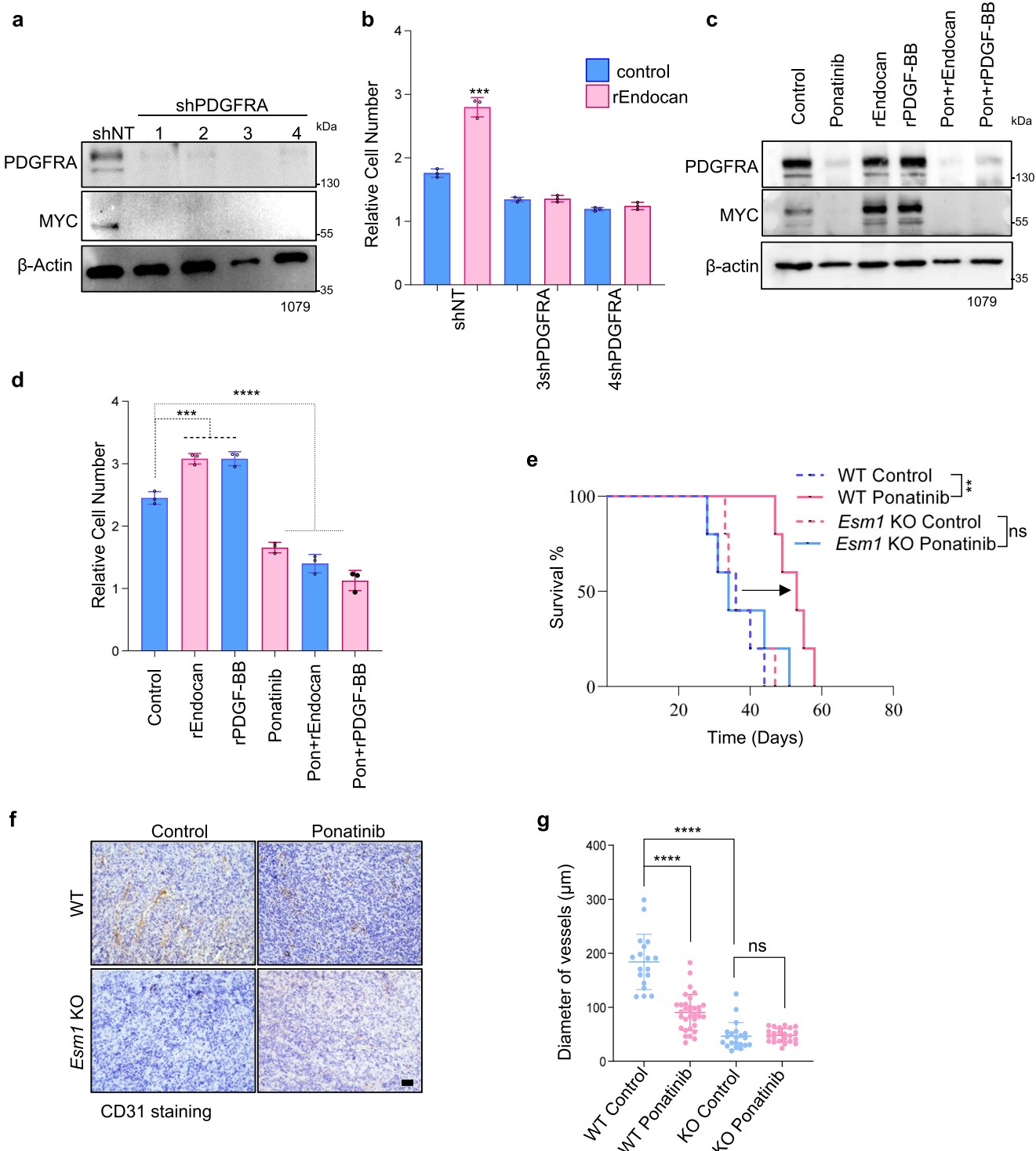

**Fig. 6 | Inhibition of PDGFRA pathway attenuates effect of Endocan on GBM cells. a** Western blot analysis of 1079 GBM cells after lentiviral mediated knockdown of PDGFRA with 4 different shRNAs against PDGFRA or Non Target shRNA (NT) as a control. Representative result of $n = 3$ biological replicates. **b** Effect of rEndocan (10 ng/ml) on proliferation of 1079 cells infected with shNT or shPDGFRA lentiviruses. $n = 3$ biological replicates. ***$P = 0.0004$ for shNT (rEndocan vs. Control*), $P = 0.7498$ for 3shPDGFRA (rEndocan vs. Control) and $P = 0.2366$ for 4shPDGFRA (rEndocan vs. Control), unpaired two-tailed Student's $t$ test. **c** Western blot analysis of 1079 cells depicting PDGFRA, MYC, and β-actin protein level expression in cells treated with rEndocan (10 ng/ml), rPDGF-BB (10 ng/ml), ponatinib (1 μM) and their combinations for 72 h, Representative result of $n = 3$ biological replicates. **d** Effect of ponatinib(1μM) on proliferation of 1079 cells incubated with rEndocan or rPDGF-BB (10 ng/ml). $n = 3$ biological replicates, one-

way ANOVA followed by Dunnett's multiple comparisons test. ***$P = 0.0001$ and ****$P < 0.0001$. **e** Kaplan–Meier survival analysis of Bl/6 mice intracranially injected with WTD and KOD cells and treated with 50 mg/kg/day dose of ponatinib for 5 days using oral gavage. $n = 5$ mice per group, $P$ value was determined by log-rank test; **$P = 0.00184$ for WT control vs. WT ponatinib, $P = 0.87$ for *Esm1* KO control vs. *Esm1* KO. Ponatinib. **f** Representative microscopic images of immunohistochemical (IHC) staining with anti-CD31 antibody of tumor sections obtained from mice as in "E". **g** Blood vessels density ($n = 18$ vessels/group) from tumor sections ($n = 3$ mice) stained with CD31 antibody was analyzed using ImageJ software. ****$P < 0.0001$ for WT Ponatinib vs. WT Control and $P = 0.7852$ (not significant) for KO Ponatinib vs. KO Control. ****$P < 0.0001$ for WT control vs. KO Control groups. one-way ANOVA followed by Dunnett's multiple comparisons test. Scale bar, 100 μm. Source data are available in the Source Data file.

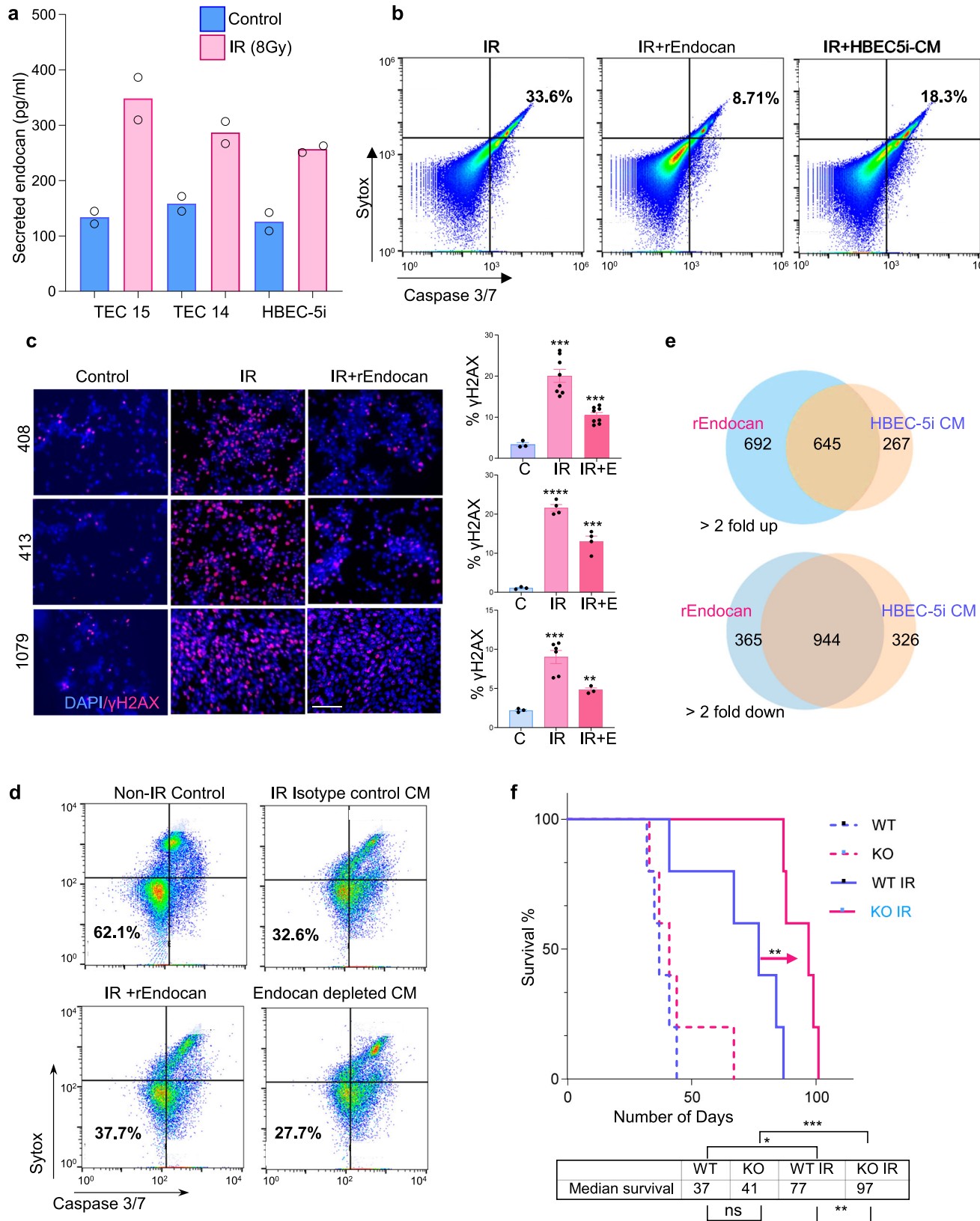

generated during our previous work allowed us to identify Endocan (*ESM1* gene) as the most significantly upregulated secreted protein in tumor-associated VE cells when compared to the normal brain endothelium. Our experiments revealed that on the one hand, Endocan affects the phenotype of GBM cells, and on the other hand, by direct and indirect mechanisms further promotes microvascular

proliferation within the tumor (Fig. 8). Using *Esm1* KO mice, we demonstrated that the absence of Endocan has a dramatic effect on both GBM cells and tumor-associated vasculature, suggesting that this protein might be one of the master regulators of GBM-VE crosstalk. One intriguing mechanistic explanation of these results can be based on the recent study from Pan and colleagues who demonstrated that

**Fig. 7 | Endocan protects glioblastoma cells from radiotherapy. a** ELISA assay comparing levels of secreted Endocan in conditioned media collected from HBEC-5i cells, TEC14, and TEC15 cells following irradiation with 8 Gy at 48-h post radiation time-point. n = 2 biological replicates. **b** Flow cytometry analysis of 711 cells treated with rEndocan (10 ng/ml) or HBEC-5i CM and for 3 days and subsequently irradiated with 8 Gy. Samples were collected on Day 5 after irradiation and stained for caspase 3/7 and SYTOX, n = 3 biological replicates. **c** Representative microscopic images of GBM cells (408, 413, 1079) treated with rEndocan (10 ng/ml) for 3 days and then treated with 8 Gy. Sham irradiated cells were used as control. Cells were fixed 16 h after radiation and stained for γ-H2AX (Ser139) (left panel). Images were quantified using n = 3 biological replicates. % γH2AX positive cells were measured by dividing γH2AX cells by a total number of nuclei (DAPI positive cells) using ImageJ software (right panel). Scalebar: 125 μm. For 408 cells, ****P = 0.0002 (IR vs. Control), ***P = 0.0002 (IR+rEndocan vs. Control). For 413 cells, ****P < 0.0001 (IR vs. Control), ***P = 0.0007 (IR+rEndocan vs. Control). For 1079 cells, ****P = 0.0008 (IR vs.

Control), ***P = 0.0012 (IR+rEndocan vs. control), unpaired two-tailed Student's t test. **d** FACS analysis of Caspase 3/7 and SYTOX staining in 1051 GBM cell pre-incubated for 3 days with HBEC-5i CM that was incubated with Endocan blocking antibody (Endocan depleted CM), or isotype control antibody; cells preincubated with rEndocan was used as a positive control. After incubation cells were irradiated at 8 Gy and analyzed after 2 more days. **e** Venn diagram representing differentially expressed genes in 1051 GBM cells irradiated with 8 Gy alone (control) or in a presence of rEndocan or CM from HBEC5i cells. Cells were collected 5 days after radiation. Genes whose expression was altered at least 2 folds were used for the analysis. **f** Kaplan−Meier survival analysis of *Esm1* WT or *Esm1* KO mice intracranially injected with 7080 cells and subsequently irradiated with 3 doses of 2.5 Gy 14 days after injection. n = 5 mice per group. P value was determined by log-rank test, ns-P = 0.35 non-significant (WT vs. KO);*P = 0.00875 (WT vs. WT IR); **P = 0.00443 (WT IR vs. KO IR); ***P = 0.00206 (KO vs. KO IR). Source data are available in the Source Data file.

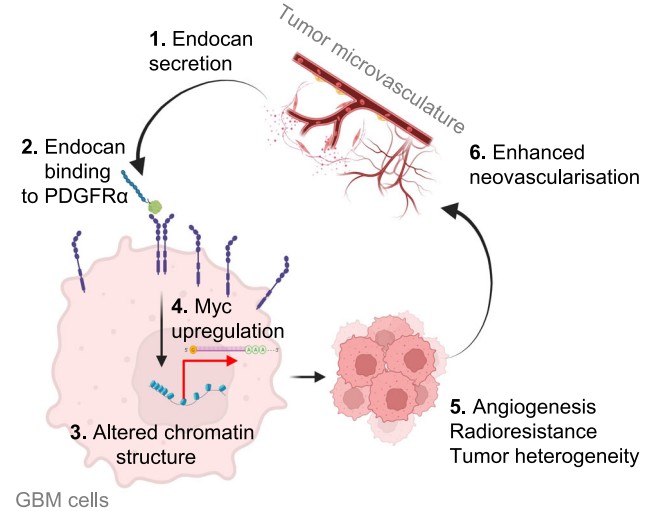

**Fig. 8 | Schematic summary illustrating the role of Endocan in GBM-VE crosstalk.** Endocan secreted by VE cells binds to PDGFRA receptor and activates downstream pathway, resulting in upregulated MYC expression and subsequent increase in GBM proliferation, radioresistance, intratumoral heterogeneity and further enhancement of angiogenesis. The figure was created in BioRender. Kornblum, H. (2024) BioRender.com/z54s425.

after translation, a fraction of Endocan molecules is not exported from the cells and instead transported to the nucleolus where Endocan activates β-catenin signaling[57] which was shown to promote angiogenesis in the postnatal brain[58]. Therefore, it is possible that VE-secreted extracellular Endocan promotes malignancy of GBM cells, while intracellular Endocan affects VE cells and enhances tumor vascularization.

Multiple prior studies have implicated Endocan in modulating cell proliferation, migration, and invasion in renal, colorectal, non-small cell lung, and hepatocellular cancers[10], and showed that its expression correlated with poor prognosis in GBM patients[7]. To determine the molecular mechanism by which Endocan affects GBM, we began by identifying its receptor on the surface of GBM cells. We demonstrated that Endocan binds to and activates PDGFRA which subsequently phosphorylates multiple downstream targets such as PI3K and ERK[59]. Endocan was previously shown to interact with CD11a integrin in human lymphocytes and Jurkat cells[60] and that it could also activate EGFR signaling in non-small cell lung cancer[46]. However, our analysis did not identify either EGFR or CD11a as Endocan binding partners in GBM. It is possible that due to its unique structure containing both an

N-terminal 18 kDa protein component and C-terminal 30 kDa dermatan sulfate chain, Endocan can interact with a high variety of cell surface receptors as well as extracellular matrix components[13]. The net functional outcome of Endocan will therefore depend on the expression level of its binding partners in addition to its relative affinity for them. This hypothesis is in good agreement with our data indicating that Endocan has a similar but not identical effect as PDGF-BB− one of the cognate ligands of PDGFRA, suggesting that Endocan may activate other yet unknown receptors in GBM. However, as we demonstrated that PDGFRA inhibition is sufficient to abrogate the effect of Endocan on GBM cells, we can speculate that, at least in our experimental model, PDGFRA is the main receptor for Endocan. Importantly, these data also provide insights into the non-canonical mechanisms of PDGFRA activation. Although the classical ligands for PDGFRs are members of PDGF family, other ligands for these receptors such as VEGF-A[61] and CTGF[62] have been identified. Therefore, Endocan may serve as an additional PDGFR ligand allowing cancer cells to activate PDGFRA signaling pathway, even if PDGFs are absent or diminished.

Interestingly, we found that ESM1 and PDGFB are expressed in the same tumor regions, while PDGFA and PDGFC seem to be mutually exclusive with PDGFB and ESM1. Consistent with this result, we showed that Endocan mimics many of the known functional capabilities of PDGF-BB as prior studies have shown PDGF-BB as a key regulator of GSC self-renewal similar to Endocan's pro-proliferative and tumorigenic properties[63,64]. In regard to vascular remodeling, PDGF-BB, much like Endocan, is highly expressed in endothelial tip cells during neoangiogenesis, regulates vessel formation[65] and permeability thereby contributing to leaky tumor vasculature and enhanced tumor growth[66–69]. These findings suggest a potential combinatorial relationship between PDGFR ligands and Endocan in GBM. It will be of great interest to determine differences in the effects of various PDGFR ligands on glioblastoma cells and to identify cells secreting these ligands within the tumor.

As most of our experiments demonstrated that Endocan promotes aggressiveness of GBM cells, one could expect that GBM tumors developed in the presence of Endocan would show more malignant properties than the Endocan-deprived counterpart. Surprisingly, this was not the case. We demonstrated that GBM cells propagated in *Esm1* KO mice were as aggressive as the ones in *Esm1* WT. Nevertheless, our further experiments revealed that Endocan does increase the malignancy of GBM if cancer cells propagated in ESM1 WT and KO mice are co-injected together into the same animal. These findings are in good agreement with previous work indicating that a higher level of GBM intratumoral heterogeneity, rather than any specific tumor phenotype, correlates with worse patient outcome[70]. We can speculate that Endocan secretion restricted to the perivascular niche enhances intratumoral heterogeneity by promoting the development of two phenotypically distinct cell populations that grow in the presence or

absence of Endocan. In our previous studies, we demonstrated that a variety of factors in the tumor necrotic core microenvironment promote mesenchymal-like transformation of GBM cells[32,71]. It is interesting to propose that the opposite process may exist in regions of microvascular proliferation, where Endocan protects GBM cells from mesenchymal differentiation and stimulates more proneural-like properties. This hypothesis is indirectly confirmed by the observation that Myc signaling is upregulated in proneural GBM tumors, while mesenchymal tumors exploit pathways related to inflammatory response and IL6/JAK/STAT3[72]. Therefore, we can argue that in different GBM regions various microenvironmental factors may differentially affect GBM phenotype, thereby promoting intratumoral heterogeneity and increasing GBM malignancy, and that Endocan protein plays an important role in this process.

From a therapeutic perspective, blockade of Endocan signaling in GBM cells can be achieved by at least four different approaches: (i) downregulation of Endocan secretion in VE cells; (ii) removal of the already secreted Endocan protein from the extracellular space; (iii) direct inhibition of PDGFRA; and (iv) inhibition of Myc. Importantly, we demonstrated that treatment with a PDGFRA inhibitor, ponatinib, more substantially increased survival of animals from the *Esm1* wild-type group as opposed to the *Esm1*-KO mice, indicating that tumors that have a high level of expression of Endocan may be more sensitive to this type of treatment. On the other hand, we showed that the removal of Endocan from conditioned medium by antibodies increases the efficacy of radiotherapy in vitro. Consistent with these latter findings, a prior study demonstrated that treatment of mice with metastatic breast cancer tumors with 1-2B7, a high affinity anti-Esm1 monoclonal antibody, yielded an improved response to bevacizumab treatment in vivo[73]. In an alternate approach, the addition of sunitinib (multi-tyrosine kinase inhibitor) to HUVEC endothelial cells has been recently shown to specifically prevent upregulation of Endocan, demonstrating another means to inhibit Endocan-tumor interactions. Finally, although no clinically successful small molecule Myc inhibitors have been developed thus far[74] multiple approaches such as blocking peptides[75] and exosome mediate knockdown[76] have shown promising results in in vivo GBM models. It is important to note that global loss of *Esm1* does not cause a noticeable defect in the brain or other organs, so systematically diminishing or even eliminating tumor-associated Endocan could indeed be a promising strategy to control glioblastoma progression and recurrence.

Unfortunately, development of resistance to Endocan-targeting therapy can be expected. Thus, we demonstrated that in the absence of Endocan, GBM cells undergo reprogramming that can desensitize them to PDGFRA inhibition. This coincides well with a recently published study that showed that upon blockade of the PDGFR signaling pathway, GBM cells acquire stable and persistent phenotypic alterations allowing them to rely on PDGF-independent proliferation mechanisms[77]. Therefore, it is important to develop approaches targeting both Endocan-PDGFRA dependent and independent populations of GBM cells within the tumor.

Although we used multiple methods and models to confirm that endothelial cell-derived Endocan can interact with PDGFRa and activate downstream signaling pathways in GBM cells, our work has several important limitations. First, it is not clear how Endocan that shares no structural similarity with classical PDGFRa ligands interacts with this receptor. To address this question, the structure of the PDGFRa-Endocan complex will need to be established. Second, it will be important to determine the binding constant of the native (correctly glycosylated) Endocan with PDGFRa dimers that are present on the surface of GBM cells. Third, differences in the effects of Endocan and PDGF-BB as well as the results of our competition assay (Fig. 4b) suggest that in addition to PDGFRa there might be other receptors for Endocan on the surface of GBM cells which have to be determined. Finally, it is still not clear by what mechanism Endocan-induced

PDGFRa activation affects the chromatin structure of the Myc promoter region.

In summary, the findings presented here provide insights into the mechanisms of intercellular crosstalk between GBM and VE. We found that the endothelial secreted Endocan protein has a strong effect on both VE and GBM cells and seems to play an important role in GBM progression. However, similar to the results earlier obtained in a clinical trial of the anti-angiogenic drug bevacizumab which failed to improve overall survival of GBM patients due to metabolic reprograming[78], glioblastoma cells can easily adapt to the absence of Endocan by using alternative signaling pathways. Therefore, in order to achieve progress in the therapy of this devastating disease, it will be critical to reveal the molecular mechanisms underlying the high plasticity and adaptation capacity of GBM cells. Our results point to the alteration in chromatin structure of Myc as one of pathways possibly responsible for such adaptation.

## Methods

### Ethics
Our research complies with all relevant ethical regulations and guidelines. This study was conducted under protocols approved by the IRBs and IACUCs of the University of California Los Angeles (UCLA), MD Anderson Cancer Center (MDA), and the University of Alabama at Birmingham (UAB). All patients were provided written informed consent for all clinical information, treatments, and prospective biopsy acquisition.

### Cell cultures
Human IDH wildtype gliomasphere cultures used in this study (patient-derived glioma sphere lines 157, 1051, 1079, 711 and 413) were maintained in DMEM/F12 medium supplemented with 2% B27 supplement (% vol), 20 ng/ml bFGF, and 20 ng/ml EGF. The bFGF and EGF were added twice a week, and the culture medium was replaced every 7 days. Experiments with neurospheres/gliomaspheres were performed with lines that were cultured for fewer than 30 passages since their initial establishment. To obtain Human Tumor-Associated Endothelial cell culture glioblastoma tissues from patients were freshly isolated during surgery, manually dissociated into single cells, and subsequently sorted for CD31+ cells using magnetic beads (ThermoFisher Scientific). Following confirmation of CD31 expression, VE cells were grown in fibronectin-coated flasks with Endothelial cell growth media (ScienCell) containing 2% FBS, 1% Penicillin-Streptomycin solution and 1% Endothelial growth supplement (ScienCell). HBEC-5i (ATCC) were cultivated in DMEM/F12 medium containing 10% FBS, 1% Penicillin-Streptomycin solution and 1% Endothelial cell growth supplement. Expression of CD31 was periodically checked by flow cytometry.

To assess PDGFRA/EGFR phosphorylation and subsequent downstream signaling cells (patient-derived glioma sphere lines or mouse GBM lines (7080)) were incubated for 24 h in media without growth factors, next recombinant Endocan/PDGF-BB/EGF was added to the cultural medium at the indicated concentrations. After that, cells were incubated at 37 °C for the indicated period of time, washed with ice cold PBS and used for the western blot analysis. STR analysis was performed to confirm cell identity (Supplementary Data Table 5). The cell lines were tested negative for mycoplasma contamination.

### Animal strains, housing and care
*Esm1* KO (C57BL/6J) mice are a generous gift from Dr. Ralf H. Adams[13]. BL6 (C57BL/6J) and NOD SCID (Prkdc^scid) mice were purchased from The Jackson Laboratory. Mice were housed in a germ-free colony animal facility ($n = 5$ mice per cage) under inspection by ARC and were observed daily. Temperature and humidity were monitored and recorded on a daily basis with temperature maintained at 68–79 °F and humidity maintained at 30–70%. The mice were kept with a 12-h

light/dark cycle. All animals were maintained in accordance with the National Institute of Health (NIH) Guide for the care and Use of Laboratory Animals and were handled according to protocols approved by the University of Alabama at Birmingham Institutional Animal Care and Use Committee and the University of California Los Angeles Animal Research Committee (ARC). The genomic status of *Esm1* was confirmed by PCR according to the prior studies. Only female mice were used in the study.

### In vivo syngeneic intracranial tumor models
Spontaneous tumors generated in adult *PTEN*, *TP53*, *NF1* deleted mice[22] were a kind gift from the Parada lab who first cultured cells as tumorspheres prior to transfer to us where cultures were propagated in neurosphere media as described previously[14]. Cultured cells were dissociated in Accutase prior to transplantation. The PDGFB-overexpressing mouse GBM model was kindly shared by Dr. Dolores Hambardzyuman (Emory University). These tumors were induced by RCAS-PDGAB lentivirus injection into Nestin-Tva/Cdkn2a-/- mice[23]. Once mice exhibited neurological symptoms (aged 6–10 weeks), they were euthanized, and the tumor was isolated and dissociated into single cells using previously published method[14].

Our general protocol for the intracranial tumor models was described previously[14]. Briefly, $5 \times 10^5$ glioblastoma cells were injected into the brains of Bl/6 or *Esm1* KO mice (Mice aged 6–8-week-old were used for experiments). Following injections, mice were monitored daily for signs of tumor burden and were euthanized following established protocols upon the observation of endpoint symptoms, including head tilt, lethargy, seizures, and significant weight loss. Our general protocol for the intracranial tumor models was described previously[14]. For immunohistochemical studies, mice were perfused with ice-cold PBS, followed by 4% paraformaldehyde (PFA). Mice brains were dissected and fixed in 4% PFA solution for 48 h and then transferred to 10% formalin for 48 h. For tumor collection to perform RNA extraction, mice were sacrificed, and tumors were isolated and fast-frozen using liquid nitrogen.

### In vitro and in vivo irradiation
Cells or mice were irradiated at room temperature using X-ray irradiator (Gulmay Medical Inc., Atlanta, GA) at a dose rate of 5.519 Gy/min for the time required to apply an 8 Gy dose. The X-ray beam was calibrated using NIST-traceable dosimetry and operated at 300 kV and hardened using a 4 mm Be, a 3 mm AI, and a 1.5 mm Cu filter. Mice were anesthetized before the irradiation. The body was covered with a lead shield to avoid whole-body irradiation.

### In vitro drug treatments
Cells were treated with ponatinib, olaratumab, or nintedanib (Selleckchem) at different time points followed by collection for Western blotting analysis or RNA extraction.

### In vivo drug treatments
Mice were injected with 7080 cells to form intracranial tumors. On day 14 post-injection, mice were given daily administration of 50 mg/kg/day of Ponatinib (Selleckchem) by oral gavage for 5 days.

### Endocan enzyme-linked immunosorbent assay (ELISA)
The concentration of secreted Endocan was measured in conditioned medium using the Endocan ELISA kit (Boster Bio) according to the manufacturer's protocol.

### Protein labeling
Recombinant Endocan and PDGF-BB were labeled with Alexa Fluor 488 Microscale Protein Labeling Kit (Thermo Fisher Scientific) and Alexa

Fluor™ 647 Microscale Protein Labeling Kit (Thermo Fisher Scientific) respectively according to the manufacturer's protocol. Labeled proteins were added to glioblastoma cells in different concentrations and after 30 min incubation on ice, cells were washed twice with PBS and analyzed by Attune NxT Flow Cytometer (Thermo Fisher Scientific). The obtained data were processed with FlowJo 10 software.

### Recombinant protein pull-down assay
To obtain a protein complex, recombinant His-tagged Endocan (R&D) was immobilized on 50 µl of HisPur Ni-NTA Magnetic Beads (Thermo Fisher Scientific) according to the manufacturer's protocol. The beads were washed 3 times with PBS and incubated for 2 h with Fc-tagged recombinant PDGFRA (R&D) under constant agitation. Next, beads were washed 3 times with PBS and bounded proteins were eluted with 300 mM imidazole in PBS and subjected to subsequent western blot analysis.

### Proximal ligation assay
Patient-derived 1079 glioblastoma cells were plated in Lab-Tek II chamber wells pre-coated with laminin. The next day, cells were treated with rEndocan (10 ng/ml) or rPDGF-BB (10 ng/ml) for 90 min, washed 3 times with ice cold PBS and fixed with 4% PFA in PBS for 15 min at room temperature. Next, cells were washed 2 times with PBS and permeabilized with 0.2% Triton-X100 in PBS for 15 min. Subsequent procedures were performed using "Duolink In Situ Orange Starter Kit" (Duolink) according to the manufacturer's protocol. Dual recognition primary antibody pair set (anti-PDGFRA and anti-p-PDGFRA) validated for PLA analysis was used (#DP0013, Abnova).

### Mass spectrometry
Recombinant His-tagged Endocan (R&D) was immobilized on 50 µl of HisPur Ni-NTA Magnetic Beads (Thermo Fisher Scientific) according to the manufacturer's protocol. The beads were washed 3 times with PBS and incubated with the fraction of plasma membrane proteins that were isolated from g1079 cells as described previously[79] and solubilized in lysis buffer (50 mM Tris-HCl, 150 mM NaCl, 1% Triton X100, 0.1% sodium deoxycholate, protease inhibitor cocktail, pH 7.5). Next, beads were washed once with lysis buffer and 3 times with PBS and bounded proteins were eluted with buffer containing 8 M Urea, 2 M Thiourea, 10 mM Tris (pH = 8). Protein concentrations were determined using the QuickStart Bradford protein assay (Bio-Rad) according to the manufacturer's protocol. Eluates from immunoprecipitations were subsequently incubated with 5 mM DTT at RT for 40 min. Then proteins were alkylated with 10 mM Iodoacetamide at RT for 20 min in the dark. Alkylated samples were diluted by the addition of 50 mM Ammonium Bicarbonate solution at a ratio of 1:4. Next, trypsin (0.01 µg per 1 µg of protein) was added, and the samples were incubated at 37 °C for 14 h. After 14 h the reaction was stopped by the addition of Formic acid to the final concentration of 5%. The tryptic peptides were desalted using SDB-RPS membrane (Sigma), vacuum-dried, and stored at −80 °C before LC-MS/MS analysis. Prior to LC-MS/MS analysis samples were redissolved in 5% ACN with 0.1% TFA solution and sonicated.

Proteomic analysis was carried out on a TripleTOF 5600+ mass spectrometer with a NanoSpray III ion source (AB Sciex, Framingham, MA) coupled with a NanoLC Ultra 2D+ nano-HPLC system (Eksigent, Dublin, CA). The HPLC system was configured in trap-elute mode. For sample loading buffer and buffer A, a mixture of 98.9% water, 1% methanol, 0.1% Formic acid (v/v) was used. Buffer B was 99.9% Acetonitrile and 0.1% Formic acid (v/v). Samples were loaded on a Chrom XP C18 trap column (3 µm, 120 Å, 350 µm × 0.5 mm; Eksigent) at a flow rate of 3 µl/min for 10 min and eluted through a 3C18-CL-120 separation column (3 µm, 120 Å, 75 µm × 150 mm; Eksigent) at a flow rate of 300 nl/min. The gradient was from 5% to 40% buffer B in 90 min

followed by 10 min at 95% buffer B and 20 min of reequilibration with 5% buffer B. Between different samples, two blank 45-min runs consisting of 5 to 8 min waves (5% B, 95%, 95%, 5%) were required to wash the system and to prevent carryover. The information-dependent mass-spectrometer experiment included one survey MS1 scan followed by 50 dependent MS2 scans. MS1 acquisition parameters were as follows: the mass range for MS2 analysis was 300–1250 m/z, and the signal accumulation time was 250 ms. Ions for MS2 analysis were selected on the basis of intensity with a threshold of 200 counts per second and a charge state from 2 to 5. MS2 acquisition parameters were as follows: the resolution of the quadrupole was set to UNIT (0.7 Da), the measurement mass range was 200–1800 m/z, and the signal accumulation time was 50 ms for each parent ion. Collision-activated dissociation was performed with nitrogen gas with the collision energy ramped from 25 to 55 V within the signal accumulation time of 50 ms. Analyzed parent ions were sent to the dynamic exclusion list for 15 s in order to get an MS2 spectra at the chromatographic peak apex. β-Galactosidase tryptic solution (20 fmol) was run with a 15-min gradient (5% to 25% buffer B) every two samples and between sample sets to calibrate the mass spectrometer and to control the overall system performance, stability, and reproducibility.

Raw LC-MS/MS data from TripleTOF 5600+ mass spectrometer were converted to.mgf peaklists with ProteinPilot (version 4.5). For this procedure, we ran ProteinPilot in identification mode with the following parameters: Cys alkylation by iodoacetamide, trypsin digestion, Triple-TOF 5600 instrument, and thorough I.D. search with a detected protein threshold of 95.0% against UniProt human protein knowledgebase. For thorough protein identification, the generated peak lists were searched with MASCOT (version 2.5.1) and X! Tandem (ALANINE, 2017.02.01) search engines against UniProt human protein knowledgebase with the concatenated reverse decoy dataset. The precursor and fragment mass tolerance were set at 20 ppm and 0.04 Da, respectively. Database-searching parameters included the following: tryptic digestion with one possible missed cleavage, static modification for carbamidomethyl (C), and dynamic/flexible modifications for oxidation (M). For X! Tandem we also selected parameters that allowed a quick check for protein N-terminal residue acetylation, peptide N-terminal glutamine ammonia loss, or peptide N-terminal glutamic acid water loss. Result files were submitted to Scaffold 4 software (version 4.0.7) for validation and meta-analysis. We used the local false discovery rate scoring algorithm with standard experiment-wide protein grouping. For the evaluation of peptide and protein hits, a false discovery rate of 5% was selected for both. False positive identifications were based on reverse database analysis. We also set protein annotation preferences in Scaffold to highlight Swiss-Prot accessions among others in protein groups.

## In vitro migration experiment
Patient-derived Glioma cells (1079, 413, 408) were used for encapsulation. Approximately 600k cells per well were seeded into Aggrewell well plates (Stemcell Technologies) one day prior to encapsulation. Next day, the spheroids were encapsulated in hydrogels using a previously described hydrogel fabrication protocol[7]. Cell migration was observed at Day 1, 3, 6, and 9 post encapsulations by acquiring phase contrast images on a Zeiss Axio.Z1 Observer microscope with a Hamamatsu Orca Flash 4.0 V2 Digital CMOS Camera and Zeiss ZEN 2 (Blue Edition) software.

## Transmission electron microscopy
Tissue from tumor-bearing mice were resected after anesthetizing and perfused with 1X PBS. Tissues were fixed in 4% paraformaldehyde and 2% glutaraldehyde. Next, the tissues were fixed with 1% Osmium tetroxide followed by dehydration in different concentrations of ethanol, and resin embedding. Following the embedding, sections were cut to 60–90 nm thickness, and then were counterstained with Uranyl Acetate and Lead citrate for imaging and analysis. Hitachi® H-7500 electron microscope was used for imaging and images were captured with NanoSprint12 AMT® camera. The outer-inner blood vessel ratio was calculated based on the inner-to-outer diameter of the blood vessel wall. The previously published Shape adjusted ellipse (SAE) method was used to calculate the diameters[25].

## ATAC sequencing and data analysis
Freshly isolated WTD and KOD tumor cells were processed for ATAC sequencing as previously described[54]. Briefly, 50,000 cells were washed with 50 mL ice cold PBS and re-suspended in 50 mL lysis buffer (10 mM Tris-HCl pH 7.4, 10 mM NaCl, 3 mM MgCl₂, 0.2% (v/v) IGEPAL CA-630). The suspension was centrifuged at 500 g for 10 min at 4 degrees. Samples were added with 50 mL trans-position reaction mix of Nextera DNA library preparation kit (FC-121-1031, Illumina). DNA was amplified by PCR and incubated at 37 °C for 30 min. MinElute Lit (Qiagen) was used to isolate DNA. NextSeq 500 High Output Kit v2 (150 cycle, FC-404-2002, Illumina) was used to sequence ATAC library. The quality assessment of the initial sequence data was conducted utilizing FastQC v0.11.9 in conjunction with multiQC v1.10.1. Paired-end reads were trimmed with Trimmomatic v0.39, employing parameters (SLIDINGWINDOW:4:15, MINLEN:35). Following trimming, the reads were aligned to the mouse genome assembly GRCm38.p6 using Bowtie2 v2.3.5.1, with the specified parameters '--very-sensitive -k 5'. Subsequently, unmapped reads and those corresponding to mitochondrial DNA were excluded from further analysis. PCR duplicates were removed utilizing sambamba v0.6.8. The resultant bam files were sorted and indexed using samtools v1.1. ATAC-Seq peak calling was performed using Genrich v0.5 under the ATAC-Seq mode. Annotation of the identified peaks were performed using the R package ChIPseeker. If a peak overlaps with the promoter region (-1000bp, +1000 bp) of any transcription start site (TSS), it is annotated as a promoter peak of the gene. If a peak does not overlap with any known gene promoters or other gene-related elements it is annotated as a "Distal" peak. Peak ranges for motif analysis were written out as fasta using bedtools getfasta v2.27.1. Motif enrichment analysis was carried out with meme-chip using the Jaspar 2024 database. Coverage tracks in bigwig format were generated using the deepTools package v3.5.1.

## RNA sequencing
cDNA were used in the library preparation using Ovation® Ultralow Library Systems (NuGEN) and samples were sequenced using an Illumina HiSeq 2000 sequencer (Illumina, San Diego, CA) in high output mode across 9 lanes of 50bppaired-end sequencing, corresponding to 4.3 samples per lane and yielding ~45 million reads per sample. Additional QC was performed after the alignmentTotal counts of read-fragments that were aligned to all the candidate gene regions were derived using HTSeq program (www.huber.embl.de/users/anders/HTSeq/doc/overview.html) with Human Hg38 (Dec. 2014) RefSeq (refFlat table) as a reference and used as a basis for the quantification of gene expression. Only uniquely mapped reads were used for subsequent analyses. Differential expression analysis was conducted with R-project and the Bioconductor package edgeR. Statistical significance of the differential expression, expressed as $Log_2$ Fold Change (logFC), was determined, using tag-wise dispersion estimation, at p-Value of <0.005 unless stated otherwise. FPKM values were reported as a measure of relative expression units.

## Spatial transcriptomics data analysis
Visium glioblastoma datsets were obtained using the SPATAData R package (https://www.cell.com/cancer-cell/pdfExtended/S1535-6108(22)00220-3). Subsequently, SPATA2 objects were converted to Seurat objects using the SPATA2 R package. Normalization was performed using the SCTransform function within the Seurat R package, followed

by batch effect correction using Harmony. To assess gene abundance colocalization across the tissue section, we calculated the Kernel density for each gene using the 'ks' package in R. Pearson's Correlation was employed to measure the co-localization between the kernel densities of two genes. Pearson's Correlation calculations were restricted to tissue spots where probabilities were greater than or equal to the 2-quantiles of the kernel densities. Gene expression levels were visualized using the SPATA2 R package.

## Flow cytometry

For CD44 staining gliomaspheres were dissociated into single cells and stained with anti-CD44-APC antibody (Miltenyi Biotec) according to manufacturer's protocol. For CD133 staining, gliomaspheres were dissociated into single cells and stained with anti-CD133-FITC (Biolegend) according to manufacturer's protocol. For apoptosis assay, cells were stained with CellEvent Caspase-3/7 Green Flow Cytometry Assay Kit (ThermoFisher Scientific) according to the manufacturer's protocol. To estimate the percentages of cells in the different phases of the cell cycle, propidium iodide (PI) dye (ThermoFisher Scientific) was used. Cells were fixed in 70% ethanol and treated with ribonuclease as recommended by the manufacturer's protocol. All samples were analyzed by Attune NxT Flow Cytometer (Thermofisher Scientific) and the data were processed with FlowJo 10 software.

## Endocan blocking antibody experiment in HBEC-5i CM

For Endocan blocking experiments, HBEC-5i cells were grown without serum and 10 µl of anti-Endocan antibody (ab103590) was added to 10 ml of CM. 10 µg of control rabbit IgG was added to another 10 ml of CM. CM was incubated 2 h at room temp with slow agitation followed by incubation with 100 µl of ProtA/G magnetic beads (Thermofisher Scientific) for 1 h. CM was then spun at 100 g for 5 min and supernatant was carefully removed without disturbing the beads and subsequently filtered through 0.22 µm filter.

## Cell proliferation assay

AlamarBlue reagent (Thermo Scientific) or Cell-Titer Glo (Promega) was used to determine the cell number under various treatments. Briefly, cells were seeded at a density of 5000 cells per well in 96-well plates. AlamarBlue reagent was added into each well and fluorescence was measured (Excitation 515–565 nm, Emission 570–610 nm) using Synergy HTX multi-mode reader (BioTek). CellTiter-glo was used at 1:1 ratio and measured using a luminometer plate reader. Readings were taken on Day 0 and Day 3 or Day 5 after treatment.

## Western blot

The cell lysates were prepared in RIPA buffer containing 1% protease and 1% phosphatase inhibitor cocktail on ice. The sample protein concentrations were determined using the Bradford method. Equal amounts of protein lysates (10 µg/lane) were fractionated on a NuPAGE Novex 4%–12% Bis-Tris Protein Gel (Thermo Fisher Scientific) and transferred onto a PVDF membrane (Thermo Fisher Scientific). Subsequently, the membrane was blocked with 5% skim milk or 5% BSA for 1 h and then incubated with the appropriate antibody at 4 °C overnight. Membranes were then washed with 3X with TBST buffer and incubated with appropriate HRP-conjugated secondary antibodies (GE Healthcare) for 1 h at RT. Protein expression was visualized with an Amersham ECL Western Blot System (GE Healthcare Life Sciences). β-Actin served as a loading control. ImageJ software (NIH) was used to analyze the Western blot results. Unprocessed images of all western blots are shown in Source Data and Supplementary Figs. Information about all used antibodies is provided in the Supplementary Data Table 7.

## Immunohistofluorescence

Immunohistofluorescence (IHF) staining was performed as described previously[14]. Briefly, tumors embedded in paraffin blocks were deparaffinized using Xylene and hydrated through 100%, 95%, and 75% ethanol gradient. Antigen was retrieved using DakoCytomation target retrieval solution pH 6 (Dako). Samples were then blocked with serum-free protein block solution (Dako) and incubated with corresponding primary antibodies at 4 °C overnight. Next, slides were incubated with Alexa Flour-conjugated secondary antibody for 1 h at room temperature and mounted in Vectashield mounting medium containing DAPI (Vector Laboratories). Nikon A1 Confocal microscope (Nikon) was used to capture images.

## γ-H2AX assay

Chamber slides or wells were covered with laminin for 24 h prior to cell seeding. Cells were allowed to grow and acclimate with treatment conditions. At the indicated times, the cells were irradiated or treated. Cells were then fixed in 4% paraformaldehyde, permeabilized with 0.5% Triton-X/PBS, and stained with an anti γ-H2AX antibody (1:500) and anti-rabbit conjugated Alexa 594 secondary antibody was used along with DAPI. Plates were imaged using EVOS microscope and ImageJ was used to manually quantify positively stained cells. At least 10–15 images were taken at 20X magnification.

## Immunohistochemistry

Immunohistochemistry (IHC) staining was performed as described previously[32]. Briefly, tumors embedded in paraffin blocks were deparaffinized using Xylene and hydrated through 100%, 95%, and 75% gradient of ethanol. Slides were then microwaved in the presence of DakoCytomation target retrieval solution pH 6 (Dako). Slides were incubated with 0.3% hydrogen peroxide solution in methanol for 15 min at room temperature to inhibit internal peroxide activity. Slides were then blocked with serum-free block solution (Dako) and incubated with corresponding primary antibody overnight at 4 °C. Next, samples were incubated with EnVision+ System-HRP labeled Polymer (Dako) and visualized with DAB peroxidase substrate kit (Vector Laboratories). Images were captured using EVOS® FL inverted microscope (Thermo Fisher Scientific IHC scoring was performed using a previously described method in a blinded fashion. Vessel quantification for IHC images was measured using Vessel Quantify_IHC and IHC profiler Plugin in ImageJ, for CD31 staining in tissues[7,80].

## Lentivirus production and transduction

Lentiviruses were produced as described before[32]. Briefly, HEK293FT packaging cells (Invitrogen) were co-transfected with lentiviral vectors encoding shRNAs or GFP or mCherry and two packaging plasmids psPAX2 and pMD2.G. Growth media was changed the following day and lentiviruses-containing supernatants were harvested at 72 h after transfection and concentrated 100-fold using Lenti-X concentrator (Clontech). For infection, GBM cells were dissociated into single cells with accutase and seeded on laminin coated 6 well plates at $8 \times 10^5$ cells per well. Next day, 8 µg/ml of Polybrene (EMD Millipore) along with viral supernatant were added to GBM cells. Two days after infection, transduced cells were selected with 1 µg/ml of puromycin (Sigma) for 3 days.

## Tissue microarray

Tissue microarray consisting of 0.6-mm cores from formalin-fixed, paraffin-embedded tissue blocks were generated using patient derived glioblastoma tissue samples at the Osaka City University (Cohort #3 $n = 38$). Patient samples were de-identified prior to transfer to us. The study involving human tissue samples was conducted in accordance with guidelines approved by the Institutional Review Board (IRB) at Osaka City University ensuring ethical handling of all research participants. Informed consent was obtained from all participants prior to the use of tissue samples.

### RNA isolation and quantitative real-time PCR

mRNA was extracted and purified using the RNeasy Mini kit (Qiagen) according to the manufacturer's protocol. Nanodrop 2000 spectrophotometer was used to determine the concentration and quality of RNA. RNA (0.5–1 μg) was reverse-transcribed in cDNA using iScript Reverse Transcription Supermix (Bio-Rad) and then amplified using the following cycling conditions; 95 °C for 5 min, and then 50 cycles of 95 °C for 20 s, 60 °C for 20 s and 72 °C for 20 s. qRT-PCR was performed on StepOnePlus thermal cycler (Thermo Scientific) with SYBR Select Master Mix (Thermo Scientific). 18 s, B-actin was used as an internal control. Primer sequences are shown in Supplementary Data file 6.

### Determination of Kd by surface plasmon resonance

Proteins were immobilized on CM5 chips using a Biacore T200 instrument by amine coupling following the manufacturer's instructions. Reagents used for immobilization were (N-hydroxysuccinimide, 1-ethyl-3-(3-aminopropyl) carbodiimide hydrochloride and ethanolamine hydrochloride) that were purchased from Cytiva. HBS-EP, the analyte buffer consisted of 0.15 M NaCl, 3 mM EDTA, 0.005% (v/v) surfactant P20, and 0.01 M Hepes (pH 7.4). For the SPR competition assay with Heparin, 1 μg/ml and 10 μg/ml of Heparin (Heparin Sodium salt from porcine intestinal mucosa, 100KU, H3393, Millipore Sigma) were used. A new channel was made with hPDGFRA and the assay was run on 2 channels along with heparin. For the data collection, binding was monitored at 1-s intervals for 3 min with an analyte flow rate of 30 μl/min. Dissociation was monitored at 1-s intervals for 2 to 7 min. Sensor chips were washed with 10 mM HCL for regeneration. BIAevaluation 4.1 (https://www.cytivalifesciences.com/en/us/support/software/biacore-downloads#) software from Biacore, was used to perform data analysis. The equilibrium dissociation constant was calculated as $KD = koff/kon$ at five different ligand concentrations.

### Statistics and reproducibility

All data are presented as mean ± the standard deviation (SD). The number of replicates for each experiment is stated in the figure legend and always refer to independent biological replicates. Statistical differences between two groups were evaluated by unpaired two tailed t-test unless mentioned otherwise in the figure legend. One-way ANOVA followed by a post-hoc t-test was used in comparisons of more than two groups. Log-rank analysis was used to determine the statistical significance of Kaplan–Meier survival curves. P values less than 0.05 were deemed to be significant. Sample size (number of mice) was predetermined based of the results of our previous studies[14,33] using similar methodologies and thus no specific statistical tests (e.g., power calculations) were used in the current study to predetermine sample size. Those performing measurements on immunohistochemistry and immunofluorescence were blinded as to treatment groups or genotypes. No blinding was performed for assessment of animal survival or in vitro experiments.

### Reporting summary

Further information on research design is available in the Nature Portfolio Reporting Summary linked to this article.

### Data availability

ATACseq data of the tumors formed by 7080 cells in ESM1 WT and KO mice were deposited to the Gene Expression Omnibus (GEO) under the accession number GSE137796. RNAseq data of the tumors formed by 7080 cells in ESM1 WT and KO mice were deposited to GEO under the accession number GSE137808. RNAseq data of patient-derived glioma sphere lines (1051 and 1079) are deposited with access codes GSE277011 and GSE277014. Raw Mass spectrometry data have been deposited in ProteomeXchange via the PRIDE with the primary accession code PXD057635 https://proteomecentral.proteomexchange.org/ui?search=PXD057635. Data pertaining to Fig. 1c was obtained from previously published study[7]. Data from Single-cell sequencing experiment from Supplementary Fig. 1e were obtained from a previously published study[15] using their published tool, GBMseq.org. Visium data from a previously published paper[16] was analyzed to generate Supplementary Fig. 1f. Data related to Supplementary Figs. 1g and h were obtained from a previously published study[18]. Source data are provided with this paper.

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

## Acknowledgements

We would like to express our sincere appreciation to all the patients and families who kindly allowed us to obtain their tumor samples for this study. We thank Dr. Ralf Adams (Max Planck Institute for Molecular Biomedicine) for generously sharing cryopreserved *Esm1* embryos. We thank the UAB and UCLA pathology, microscopy, flow cytometry, and TCGB sequencing cores for their technical assistance. This work was supported by NIH grants R01NS083767, R01NS087913, R01CA183991, R01CA201402 (I.N.), R01NS104339 (A.B.H.), RO1CA241927 (S.K.S.), NS052563 (H.I.K.), the UCLA SPORE in Brain Cancer, P50 CA211015 (H.I.K.); and the Dr. Miriam and Sheldon G. Adelson Medical Research Foundation (H.I.K., A.L.B., S.A.G., and L.A.H.). S.B. was supported by the UCLA Chancellor Postdoctoral Fellowship award. K.S.A. was supported by grant 075-15-2019-1669 from the Ministry of Science and Higher Education of the Russian Federation. T.F.K. was supported by the Russian Science Foundation grant 24-15-00097.

## Author contributions

S.B., M.S.P., I.N., A.B.H., S.A.G., L.A.H., and H.I.K. conceptualized and designed the study. S.B., M.S.P., N.S., S.Y.Y., B.O., D.Y., M.S.K., Y.G., S.D.M., and J.A.O-P. performed patient sample processing, cell culture, and related in vitro assays, under the supervision of I.N., A.L.B., and H.I.K. S.B. prepared and N.P.B. performed electron microscopy study of tumor samples under the supervision of L.A.H. G.J. performed BiaCore/ SPR analysis under the supervision of T.G. S.B., and A.S., performed 3D hydrogel studies under the supervision of S.K.S. and H.I.K. S.B. and N.S. performed mouse in vivo intracranial surgeries, and in vivo drug testing, under the supervision of I.N., A.B.H., and H.I.K. S.B. prepared samples for RNA sequencing and ATAC sequencing. K.S.A., T.F.K., R.K., and Y.Q. performed Bioinformatics analysis. S.B., M.S.P., and N.S. conducted a formal analysis of the study under supervision of I.N. and H.I.K. S.B. conceptualized and wrote the first draft of this paper under the supervision of I.N. and H.I.K. S.B., M.S.P., N.S., M.A.N., I.N., and H.I.K. reviewed and edited the paper. All authors reviewed and approved the final version of the paper. M.S.P., K.S.A., A.B.H., L.A.H., A.L.B., S.A.G., I.N., and H.I.K. acquired funding for the study.

## Competing interests

The authors declare no competing interests.

## Additional information

[1]The Intellectual and Developmental Disabilities Research Center, The Semel Institute for Neuroscience and Human Behavior, and The Broad Stem Cell Research Center, The Jonsson Comprehensive Cancer Center, David Geffen School of Medicine at UCLA, Los Angeles, CA, USA. [2]Shemyakin-Ovchinnikov Institute of Bioorganic Chemistry, Moscow, Russia. [3]Precision Medicine Institute, University of Alabama at Birmingham, Birmingham, AL, USA. [4]Johns Hopkins University School of Medicine, Baltimore, MD, USA. [5]Department of Bioengineering, University of Texas at Austin, Austin, TX, USA. [6]Department of Neurology, Icahn School of Medicine at Mount Sinai, New York, NY, USA. [7]Department of Neurosurgery, Ehime University Graduate School of Medicine, Shitsukawa 454, Toon, Ehime, Japan. [8]Center for Precision Genome Editing and Genetic Technologies for Biomedicine of Lopukhin Federal Research and Clinical Center of Physical-Chemical Medicine of Federal Medical Biological Agency, Moscow, Russia. [9]Lopukhin Federal Research and Clinical Center of Physical-Chemical Medicine of the Federal Medical and Biological Agency, Moscow, Russia. [10]Department of Medicine, Center for Iron Disorders, David Geffen School of Medicine at UCLA, Los Angeles, CA, USA. [11]Interdepartmental Program in Bioinformatics, Program in Neurogenetics, Department of Neurology and Department of Human Genetics, David Geffen School of Medicine at UCLA, Los Angeles, CA, USA. [12]Department of Pharmaceutical Chemistry, University of California, San Francisco, CA, USA. [13]Departments of Neurology and Neuroscience, Icahn School of Medicine at Mount Sinai, James J Peters VA Medical Center, Bronx, NY, USA. [14]Center for Translational Neuromedicine, University of Rochester Medical Center, Rochester, NY, USA. [15]Faculty of Health and Medical Sciences, University of Copenhagen, Copenhagen, Denmark. [16]Department of Cell, Developmental and Integrative Biology, University of Alabama at Birmingham, Birmingham, AL, USA. [17]Department of Neurosurgery, Harada Hospital, Iruma, Saitama, Japan. [18]These authors contributed equally: Soniya Bastola, Marat S. Pavlyukov. [19]These authors jointly supervised this work: Ichiro Nakano, Harley I. Kornblum. ✉e-mail: dr.ichiro.nakano@gmail.com; hkornblum@mednet.ucla.edu

