## [Transparent Peer Review file · Nature Communications]

Endothelial-secreted Endocan activates PDGFRA and regulates vascularity and spatial phenotype in glioblastoma

Corresponding Author: Dr Harley Kornblum

Version 0:

Reviewer comments:

Reviewer #1

(Remarks to the Author)
co-reviewed with Reviewer #2.

Reviewer #2

(Remarks to the Author)

Endocan (ESM1) is an endothelial cell secreted glycoprotein. Glioblastoma (GBM) is an aggressive cancer for which neovascularization is critical for cancer progression. Endocan expression is upregulated in several cancers and a reasonable amount of work has been done to show that ESM1 plays a role in cancer development and progression. However, how ESM1 contributes to GBM and how GBM cancer cells and ESM1 overexpressing endothelial cells in cancer microenvironment communicate during tumorigenesis and metastasis is not fully understood at the mechanistic level. In this work, the authors report that endothelial cell secreted ESM1 plays a key role in promoting angiogenesis, proliferation, and migration of GBM cells. Importantly the authors identify PDGF Receptor α (PDGFRA) on the GBM cells as a possible receptor for endothelial cell secreted ESM1, thereby a paracrine regulation of GBM propagation is established. Additionally the authors report that endocan overexpression is key in conferring radioprotection phenotype observed in GBM. Using ESM1 WT and knock-out mouse models, along with PDGFRA inhibitor ponatinib, the authors further investigate this newly discovered axis in GBM. Mechanistically, the authors report significant upregulation of MYC expression mediated by ESM1-PDGFRA interaction which they propose further promotes hyper cell proliferation in GBM.

Overall, this a very well written manuscript in which major claims are supported by extensive and convincing figures and subfigures. Use of molecular, cellular, spheroids genome-wide ATAC and animal based studies in addressing key questions on ESM1 function in GBM neovascularization and progression makes this manuscript mechanistically in depth. The work is extensive and identifies ESM1-PDGFRA axis as a potential target to understand GBM and for future translational and clinical research. Once the manuscript is sincerely revised addressing the following questions, the manuscript should be acceptable for publication.

1. Endocan/ESM1 is also dysregulated in cancer cells in addition to its observed hyperexpression in the vasculature. Do the authors find a parallel role of cancer cell derived ESM1 in GBM progression in their model?
2. ESM1 has been shown to associate with (mutant) EGFR in NSCLC and promotes proliferative activity along with EGF. Although the authors claim ESM1-PDGFRA interaction is the major pathway driving ESM1 mediated GBM phenotype, did the authors also determine whether blockade of EGFR function does not interfere with ESM1 action in their model system.
3. Do the authors find any major alteration in immunogenic response/infiltration of immune/macrophage cells in ESM1 WT and KO models of GBM?
4. Nuclear localized ESM1 influence TCF/LEF/ β Catenin transcriptional pathways in cancer cells. Do the authors find any dysregulation of β Catenin transcription program in endothelial/GBM cells? If not a discussion in this regard will be helpful and important for mechanistic purposes.
5. What is the rationale for comparing only the genes upregulated by recombinant Endocan/PDGF-BB (Fig.4E)? Surely these treatments result in down-regulated genes as well. A more comprehensive analysis of total differentially expressed gene sets in these treatments would be warranted.
6. It's unclear from the manuscript whether the ATAC-seq was performed on sphere cultures of WTD/KOD 7080 cells or from freshly isolated tumors without further culture. The results state that "...we performed ATAC-seq with Esm1 WTD and KOD

cells” while the Fig. 5 legend states “ATAC-seq analysis of tumors...”, the methods section invokes Ref.39 in which ATAC-seq was performed on gliomaspheres. Also, the ATAC-seq analysis section is incomplete. Please specify which version of the mouse genome the data is aligned against and how differential chromatin accessibility was determined. The ATAC-seq methods also state that “UCSC Genome Browser was used to determine whether open regions displayed H3K27ac and conserved TF binding sites.” How was this analysis used in this manuscript?

7. The authors identify 38 genes containing differentially accessible chromatin between WTD and KOD samples. Presumably these genes harbor differentially accessible chromatin regions within the gene or at some set distance (not specified) from the TSS. How many differentially accessible peaks were identified overall? There may be a significant number of differentially accessible intergenic sites contained in potential long-range enhancers.
8. Fig. 5A legend “distal” should be defined. Likewise, what constitutes promoter proximal should also be defined.
9. While it’s very impressive that ATAC-seq identifies open chromatin structure of MYC promoters (and, also potential enhancers/introns) in tumors developed in ESM1 WT but not ESM1 KO animals, it is surprising the genomic regions with E-box sequences (MYC sites) is not enriched (at least not reported here). How do the authors think that ESM1 signaling only creates a very small amount of chromatin alteration? Did the authors perform de novo motif analysis at differentially accessible sites? If other TF binding sequences are also enriched, the authors should report those as well. Do the authors find any significant changes in H3K27ac and H3K4me3 in cells used for ATAC-seq?
10. The spatial identity and tumor heterogeneity (as to why animal survival was significantly reduced only when both KOD or WTD cells are injected together but not alone) claims (good suggestions though) are speculative not supported by data provided and thus need to be modified.
11. Fig S4c can be moved to main figure. Also pertaining to Fig S4c, it would appear from the GSEA results, that rEndocan activates MYC targets better than rPDGF-BB. Did rPDGF-BB induce MYC mRNA in 413 cells to a similar level as rEndocan? The western blot in Fig. 6c suggests both rEndocan and rPDGF-BB can induce MYC protein to similar levels although this blot was done in 1079 cells while the RNA-seq was performed in 413 cells.
12. What happens to GBM cells when Dox inducible ESM1 is expressed? Do they become independent of endothelial cell secreted ESM1?
13. Discussion is long and can be shortened.
14. Fig. 4: It is interesting to note that while both ESM1 and PDGF-BB activates PI3K, p42 is only activated under PDGF stimulation. What’s the biological significance considering pathways controlled by p42/44.
15. The actin loading control for Fig.5f is problematic. I don’t think this represents a bona fide difference between WTD and KOD cells though since actin in Fig. 5e looks fine.
16. Figs 5G,5I, 6B, 6C, 6D, etc. legends should indicate how much rEndocan, rPDGF-BB, ponatinib were used.
17. Fig 6a: A better/cleaner blot for MYC is preferred. Are these cells treated with rESM1?
18. How does ESM1 signaling interface with IBSP (vascular enriched integrin binding sialo protein) function in GBM regarding cell proliferation, migration, and mesenchymal transition the authors reported in Cell Rep, 2022 paper. It appears that these two factors may regulate overlapping cellular phenotype.
19. Does IDH (WT/mutant) phenotype influences ESM1 function in GBM? The authors should inform the IDH status of the patient samples used and include relevant sentences in discussion.
20. Considering that radiation promotes vascular endothelial and pericyte-like phenotype in GBM, does ESM1/endocan treatment alters these cellular phenotypes?

Reviewer #3

(Remarks to the Author)

This is a beautiful study describing a novel bidirectional crosstalk mechanism between glioma cells and blood vessels with considerable translational potential. It provides a very important new foundation for the plausible concept that blood vessels are actually of high pathobiological importance for brain tumors, too - but that we have not hit the right target (mainly VEGF-A) so far.

The manuscript is very mature and complete; the data is presented very well, the statistics is sound.

I have only one single point: Fig. S7g: is there a quantification of this dataset? (If yes, please add; if not, the survival data in Fig. 7 and these exemplary images are sufficient).

Reviewer #4

(Remarks to the Author)

In this manuscript, the authors study the interactions between vascular endothelial cells (VECs) and GBM cells in the tumor microenvironment. This work was initiated owing to the fact that GBM is a highly vascularized tumor and VECs may play paracrine roles in controlling GBM cell growth, migration and invasion. Their initial findings show that VECs isolated from GBM tumors produce soluble factors to promote GBM tumor cell proliferation. Using an unbiased approach of RNAseq, they identified Endocan (ESM1, an endothelial-secreted proteoglycan) as most unregulated secretory protein produced by VECs from GBM tumors. They demonstrate that Endocan promotes GBM cell proliferation and migration and genetic deletion of Esm1 ablates these effects. In both in vitro and in vivo assays they show that Endocan has both autocrine and paracrine effects on both tumor vasculatures and tumor cells. To study the signaling pathways of Endocan, they employed both pull-down and proximal ligation assays to uncover PDGFRA as a potential receptor for Endocan. Indeed, Endocan and PDGF-BB compete the PDGFRA binding. Although both Endocan and PDGF-BB bind to the same PDGFRA, there seem some differences in their signaling events. Finally, the authors took both genetic and pharmacological approaches to demonstrate that targeting PDGFRA abolishes the Endocan effects on GBM growth.

Taken together, the authors employed multidisciplinary approaches to investigate the crosstalk between tumor vasculatures and GBM cells. These findings are interesting and clinically significant. Their data also provide novel mechanistic insights into GBM progression and invasion. I have several comments that may help authors to improve their findings:

1. It is known that Endocan is a specific marker for endothelial tip cells and its expression is unregulated in a number of tumors other than GBM. It is also known that VEGF, FGF and other cytokines upregulate Endocan production in tumor vascular endothelial cells, which is correlated with tumor hypervascularization. I assume that these known functions also exist in GBM as reported in other tumor types. On the basis of these published known effects, the authors should move the results related to these data to the Supplemental Information. In this way, their rest of findings would highlight the novelty.

2. The key surprising findings is that Endocan binds to PDGFRA and trigger the subsequent signaling cascade. These are totally unexpected findings. The authors should emphasize these key findings in the abstract, significance and elsewhere throughout the manuscript. As an expert in the field, this reviewer have a number of questions, including: a) The PDGF ligands are made as dimers through disulfide bonds. All members within the PDGF family and VEGF family belong to this type of growth factor ligands. Endocan as a heparan sulfate proteoglycan (HSPG) share absolutely no structural similarities with PDGFs. How would they explain that Endocan could even bind to and trigger dimerization of PDGFRs. I do not know any mechanisms to explain this. The only possibility of binding to PDGFRA is through the heparin binding property of endocan. PDGF-BB and other ligands also possess heparin-binding properties, which allow them to interact with the low affinity receptors (HSPGs). Would it be possible for Endocan to interact with PDGFRA through heparin binding? What happens if the authors perform an experiment by saturating PDGFRA heparin binding, followed by endocan binding? 3) If heparin binding is involved as a low affinity, it may help PDGFRA interact with its specific ligands and thus enhancing downstream signaling. In their experimental system, the PDGF ligands always present because the serum contains high concentration of PDGFs, including PDGF-AA, PDGF-BB and PDGF-AB related by platelets during coagulation. They should completely remove PDGFs in order to make this conclusion. 4) They should provide Kd of Endocan-PDGFRA binding; 5) The TKI drugs for the type of experiments is inappropriate because of their broad effects. Specific neutralizing antibodies against PDGFRA should be used.

The reason why I ask for these additional experiments to strengthen their conclusions is that their claim of Endocan in binding and triggering PDGFRA is very provocative in this research field. The authors need to provide convincing and robust experimental data to support this conclusion. I hope this would be helpful for them to improve the quality of their work.

3. The radiation therapy part should be omitted from this manuscript because of the flow and mechanistic link is too vague.

4. The authors should consider to cite the following publications:

PMID: 27608497
PMID: 18392794
PMID: 23773831

I hope that my comments are helpful for improving the quality of the authors' otherwise very important work. I look forward to reading a revised version.

Version 1:

Reviewer comments:

Reviewer #1

(co-reviewing with Reviewer #2)

Reviewer #2

(Remarks to the Author)

Comments on revised manuscript:

We feel that the authors have sincerely responded to the critiques of all reviewers, resulting in revised text and inclusion of new results and figures. The manuscript should be acceptable once the following changes are implemented.

1. Fig. 4E requires modification and explanation. The immunoblot in Fig.4E purports to show 10 total samples but only the top three blots (p-PDGFRA, PDGFRA, bActin) contain 10 lanes. The next 5 blots contain only 9 lanes.

2. Response of authors to reviewers 1 and 2: "Consistent with our recently published results (Nat Commun. 2022; 13: 6202) we detected substantial upregulations of all markers after irradiation, however rEndocan didn't show any reproducible effect on radiation induced vascular transdifferentiation under these experimental conditions. We have not included these data in the revised manuscript.

Comment: The authors should include these results as supplemental data in the manuscript as irradiation studies are included in the manuscript (Fig 7) and these results are relevant in that context.

3. Response of authors to reviewer 3: "Taken together, these results make it unlikely that the interaction of Endocan with PDGFR α is due to its heparin-binding capacity. We have not included these data in the already lengthy revised manuscript."

Comment: We believe Reply figure 11B and C need to be included in supplemental data for new Fig 4B as a key control that addresses reviewer 3's questions regarding binding specificity.

4. Discussion continues to remain extremely long.

5. It would have been easier if the authors highlighted the text in the manuscript where major changes were made.

6. Fig 8 can be included as Fig 7G.

Reviewer #3

(Remarks to the Author)

The authors have responded well to my comment.

Reviewer #5

(Remarks to the Author)

Bastola et al. "Endothelial-secreted Endocan acts as a PDGFR alpha ligand..." (NCOMMS-23-44699A)

I think this paper reports an interesting finding and that the authors overall have dealt satisfactory with the criticism from the reviewers. However, the point raised by reviewer 4 regarding the affinity by which Endocan binds to PDGFR alpha and whether it induces receptor dimerization have not been satisfactory addressed.

1. The authors have used SPR to determine the K_d of Endocan binding to PDGFR alpha. They have come up with a value of 345 nM, which they claim is in the same range as previously has been reported by Mamer et al. (2017) Sci.Rep. (420 nM for the binding of PDGF-BB to PDGFR alpha; 660 nM for the binding of PDGF-AA to PDGFR alpha), who also used SPR for the analysis. However, these values determined by SPR are very different from those determined by conventional Scatchard analyses. Thus, values of 0.2 nM for PDGF-AA and 0.5 nM for PDGF-BB binding to PDGFR alpha on living cells, were reported by Claesson-Welsh et al. (1989) PNAS 86, 4917-4921. It should be noted that the values determined by Scatchard analysis corresponds nicely to the half-maximum effect by PDGF-AA and PDGF-BB on receptor activation measured as autophosphorylation and cell proliferation. Thus, the SPR data are artifactual. A possible reason could be that on living cells with a possibility for movement of the receptors in the plasma membrane, a dimeric PDGF molecule binds to two receptors, whereas in the SPR analysis the receptors are immobilized, thus possibly preventing dimeric interactions and as a consequence giving values indicating about a thousand-fold lower affinity. The authors need to perform a regular Scatchard analyses on living cells, comparing the binding of Endocan and PDGF-BB.

2. The authors show a PLA-based analysis of receptor dimerization in response to Endocan binding, with PDGF-BB as control. However, it is not clear exactly how this analysis was done and what the values on the y-axis mean ("Dimerization intensity"). The authors need to perform a conventional cross-linking analysis, comparing dimerization in response to binding of Endocan and PDGF-BB.

3. There is also room for improvement of the immunoblots shown. In Fig. 4e, the lanes are poorly aligned, and there are several P-PDGFR α bands (also the case for other figures); markers of molecular mass should be inserted so that one can compare with the blot for PDGFR α (this goes for all immunoblots). Fig. S4b suggests that PDGF-BB causes dramatically more auto-phosphorylation than Endocan. However, quantification is difficult since the authors show only a kinetic analysis. The authors need to perform dose-response analyses, comparing the auto-phosphorylation of PDGFR alpha in response to different concentrations of Endocan and PDGF-BB.

Version 2:

Reviewer comments:

Reviewer #5

(Remarks to the Author)

In their response to my criticism, the authors argue that they have shown by many assays that Endocan activates PDGFR-alpha. I do not dispute this, and I do think their observations are interesting and should be published. However, the quantifications of the binding affinities are not satisfactory, neither in the original version nor in the revised version.

1. The SPR values of the interaction affinities are artifactual (for PDGF, about a thousand-fold lower than what has been measured by Scatchard analysis, and functional assays). The authors themselves state on page 9, line 14 that both PDGF-BB and Endocan induce phosphorylation of PDGFR-alpha at sub-nanomolar concentrations. So, they contradict themselves. I strongly recommend to remove the artifactual SPR data which only causes confusion.

2. Regarding dimerization, I am surprised that the conventional cross-linking assay did not work. Anyhow, the "dimerization experiment" that is shown in Fig. S4a, does not record dimerization. In the PLA assay, an antibody against PDGFR-alpha and another against p-PDGFR-alpha were used. Thus, the assay measures auto-phosphorylation of PDGFR-alpha, not dimerization. This should be clarified. In fact, there is no experiment shown that addresses the issue of dimerization. Moreover, the experimental condition used should be specified better in this experiment and in many others.

3. The authors now have performed a dose-response experiment and have included the result as a novel Figure 4e. The

experimental conditions are not specified, so the experiment is difficult to evaluate. Was incubation at 0 C or 37 C? I suspect that it was at 37 C, since the PDGF-BB activation of the receptor goes down at the highest concentrations of PDGF-BB. It should be noted that at 37 C, one measures the net effect of activation of auto-phosphorylation of the receptor and internalization and degradation; the latter could be quite substantial after a 30-minute incubation. Better is to perform such an experiment at 0 C, when there is no internalization and degradation. Moreover, inspection of the blot reveals that PDGF-BB induces auto-phosphorylation of the receptor at 0.15 nM, which is consistent with the expectation (assuming that it is the lower band that represents the receptor; by the way, the authors need to put in molecular mass markers in the immunoblots, in order to make it possible to evaluate them; I pointed this out in my previous report). Endocan seems to give a very faint band at 0.015 nM, but no band at 0.5 nM, and a more impressive band at 1.5 and 5 nM. It is very difficult to conclude from this experiment that Endocan activates the receptor at sub-nanomolar concentrations. The authors need to show a more conclusive experiment. The fact that different maximum levels are found for stimulation with PDGF-BB and Endocan is interesting, and may be related to differences in ligand-induced internalization, something that could easily be investigated.

Version 3:

Reviewer comments:

Reviewer #5

(Remarks to the Author)

The authors have addressed the two first points of criticism I raised in a satisfactory manner. However, the third point has not been addressed. I reiterate for the third time that the authors must put markers of molecular masses in their immunoblots; without this information, it is not possible to evaluate the blots. In their rebuttal, the authors claim that all bands represent the receptor. The band with lowest molecular mass is absent before stimulation with PDGF-BB or Endocan and increases upon ligand stimulation, whereas the two uppermost bands are seen also without stimulation with PDGF-BB or Endocan. In the absence of any molecular mass markers, I draw the conclusion that the lower band represented the receptor (maybe I was wrong). If the uppermost bands represent the full length receptor, why is there a background activation of the receptor? If so, the lower band may be a fragment of the receptor, possibly occurring upon internalization and degradation of the receptor. If the authors had followed my advice and performed the experiment on ice, it is likely that the pattern would have been easier to interpret. The authors argue that they have chosen to perform the experiment at 37 C since this is "more relevant". Well, this depends on the question asked. At 37 C there is internalization and degradation of the receptor which complicates the interpretation. If the authors insist of performing the experiment at 37 C, they could at least have chosen a shorter time of incubation, i.e. 5-10 minutes, when internalization and degradation have barely started.

The authors have a very interesting finding to report. I am confident that they do not want to pollute their excellent paper with substandard blots. I strongly recommend the author to perform the immunoblotting experiment again.

Point by point reply to the reviewers' comments

We thank the reviewers for their thoughtful and constructive comments. We are pleased that the general tone of the reviews was positive, but we acknowledge that there were several points made that could improve the manuscript. Therefore, in response to these helpful critiques we have made significant changes to the manuscript, all of which, we believe, support our overall conclusions. While each point will be addressed in detail, the most significant changes that we made are:

- 1) Using surface plasmon resonance (SPR), we determined the K_d of Endocan binding to PDGFRa and found that its affinity to PDGFRa (345 ±25 nM) is similar to that of PDGFbb (420 nM) (**Reply Figure 11A**; manuscript Figure 4B).
- 2) We analyzed a recently published spatial transcriptomic dataset of GBM tumor tissues (Neurooncol Adv. 2023, 5(1):vdad142). First, we confirmed that within a tumor *ESMI* expression is highly co-localized with *VWF* - a marker of endothelial cells (Exp Mol Pathol. 2002, 72(3):221-9) (**Reply Figure 1C-D**; manuscript Figure S1F). And, second, we unexpectedly found that *ESMI* and *PDGFB* are often expressed in the same tumor regions, while *PDGFA* and *PDGFC* seem to be mutually exclusive with *PDGFB* and *ESMI* (**Reply Figure 1C-D**; manuscript Figure S5A-B). The reason for this is unclear, but we speculate that Endocan may act together with PDGFbb to maintain a high level of PDGFRa activation in the same tumor region, while other PDGFR ligands are expressed in the different GBM areas and presumably affect other populations of GBM cells. In future studies it will be of great interest to determine if different PDGFR ligands stimulate different phenotypic changes in glioblastoma cells and also to identify cells secreting these ligands within the tumor.
- 3) We provide a more detailed description of our ATACseq analysis and used these data to perform *de novo* motif analysis which identified enrichment of binding sites of various transcription factors whose expression is regulated by MYC (**Reply Figure 5**; manuscript Figure S5C).

REVIEWER COMMENTS

Reviewer #1 (Remarks to the Author):

co-reviewed with Reviewer #2.

Reviewer #2 (Remarks to the Author):

Overall, this a very well written manuscript in which major claims are supported by extensive and convincing figures and subfigures. Use of molecular, cellular, spheroids genome-wide ATAC and animal based studies in addressing key questions on ESM1 function in GBM neovascularization and progression makes this manuscript mechanistically in depth. The work is extensive and identifies ESM1-PDGFR α axis as a potential target to understand GBM and for future translational and clinical research. Once the manuscript is sincerely revised addressing the following questions, the manuscript should be acceptable for publication.

1. Endocan/ESM1 is also dysregulated in cancer cells in addition to its observed hyperexpression in the vasculature. Do the authors find a parallel role of cancer cell derived ESM1 in GBM progression in their model?

Reply: We thank reviewer for this important comment. In the first version of the manuscript, we analyzed Endocan secretion level by ELISA and demonstrated that brain endothelial cells secrete substantial amounts of this protein, while GBM cells and normal human astrocytes produce nearly undetectable levels of Endocan (**Reply Figure 1A**; manuscript Figure 1D). To confirm this finding and to place it in the context of GBM *in situ*, we first analyzed a single cell RNAseq dataset derived from human GBM tumors (Cell Rep. 2017, 21(5):1399-1410) and showed a significant level of *ESM1* mRNA expression in endothelial cells but not in neoplastic cells, myeloid cells, neurons, oligodendrocytes, OPCs or astrocytes (**Reply Figure 1B**; manuscript Figure S1E). Next, we analyzed a recently published spatial transcriptomic dataset of GBM tumor tissues (Neurooncol Adv. 2023, 5(1):vdad142). Consistent with our initial results, we demonstrated that within the tumor, *ESM1* mRNA expression is highly colocalized with *VWF* mRNA expression - a marker of endothelial cells (Exp Mol Pathol. 2002, 72(3):221-9) (**Reply Figure 1C-D**; manuscript Figure S1F). Based on all these findings, we argue that in GBM, Endocan is primarily secreted by endothelial cells and we do not find strong evidence for secretion by GBM cells.

Reply Figure 1: **A**, ELISA assay comparing levels of secreted Endocan in conditioned medium from Normal Human Astrocytes (NHA), Human Brain Endothelial Cells (HBEC-5i), Tumor-associated Endothelial cells (TEC15, TEC14), and GBM cells (1079, 157, 711, and 1051). One-way Anova and Tukey post-hoc test, n=2 independent experiments, ****P<0.0001. **B**, Analysis of Darmanis single cell RNAseq dataset (Cell Rep. 2017, 21(5):1399-1410) showing ESM1 expression levels in different types of cells within the GBM tumor. **C**, Expression of ESM1, VWF, PDGFA, PDGFB and PDGFC in GBM samples from previously published Visium dataset (Neurooncol Adv. 2023, 5(1):vdad142) (n = 5 different patients). **D**, Quantification of ESM1 expression colocalization with the expression of VWF, PDGFA, PDGFB and PDGFC in samples as in “B”. Data points related to different patients are indicated in different colors. Wilcoxon signed rank-sum test, *P<0.05.

2. ESM1 has been shown to associate with (mutant) EGFR in NSCLC and promotes proliferative activity along with EGF. Although the authors claim ESM1-PDGFRa interaction is the major pathway driving ESM1 mediated GBM phenotype, did the authors also determine whether blockade of EGFR function does not interfere with ESM1 action in their model system.

Reply: We attempted to use EGFR inhibitors to assess the contribution of EGFR signaling to the effect of Endocan. However, we found that these inhibitors also influence PDGFRa activation and substantially decrease viability of GBM cells (not shown). Therefore, we were unable to determine if the blockade of EGFR function interfered with the effect of Endocan on GBM cells. To overcome this problem, we chose the strategy of determining whether Endocan has the capacity to activate the EGFR. We analyzed EGFR (Y1068) and PDGFRa (Y720) phosphorylation levels in 413 and 1079 gliomaspheres treated with recombinant Endocan or EGF as a positive control (**Reply Figure 2**; manuscript Figure S4C). EGF but not Endocan induces EGFR phosphorylation, consistent with the hypothesis that Endocan acts through PDGFRa but not through EGFR activation. Importantly, in this experiment we applied a 10 fold higher concentration of Endocan than was used in other experiments to assess PDGFRa activation, but even at this high concentration we didn't detect any effect of Endocan on EGFR phosphorylation. Therefore, we conclude that Endocan does not activate canonical EGFR signaling in our system. It is also interesting to mention, that in 1079 cells EGF was able to induce PDGFRa phosphorylation. This result is consistent with finding of Chakravarty and colleagues who demonstrated that in some patient-derived GBM cell lines PDGFRa and EGFR form heterodimers which can be activated by EGF (Sci Rep. 2017, 7: 9043).

Reply Figure 2: WB analysis of 413 (left) and 1079 (right) cells incubated with rEndocan (100ng/ml) or rEGF (100ng/ml) for the indicated period of time.

3. Do the authors find any major alteration in immunogenic response/infiltration of immune/macrophage cells in *Esm1* WT and KO models of GBM?

Reply: To address this important question we performed IHC staining of tumor sections obtained from *Esm1* WT and KO mice with antibodies for various markers of immune cells: *Iba1*, *Cd68* and *F4/80* (microglial/macrophage markers), and *Cd8* (T-cell marker) (Mol Psychiatry. 2018, 23(2):177-198; J Exp Med. 2005, 201(10): 1615–1625; Nat Rev Cancer. 2020, 20: 218–232). Although there was a relatively high degree of variability, we did not observe any significant differences between the tumors grown in *Esm1* WT and KO animals (**Reply Figure 3**; manuscript Figure S2E). This result indicates that the phenotypic differences between tumors isolated from WT and KO mice are unlikely to be due to altered immune cell infiltration, although we cannot rule out the possibility that there are differences in immune populations that we have not yet studied.

Reply Figure 3: Microscopic images of WTD and KOD tumor sections stained for IBA1, CD68, F4/80 and CD8. Quantification of the images was performed using samples from n = 6 different animals per group. n.s. – non-significant; unpaired two-tailed t-test.

4. Nuclear localized ESM1 influence TCF/LEF/ β Catenin transcriptional pathways in cancer cells. Do the authors find any dysregulation of β Catenin transcription program in endothelial/GBM cells? If not a discussion in this regard will be helpful and important for mechanistic purposes.

Reply: This is an important point and we thank the reviewer for bringing it to our attention. We believe that the data showed in **Reply Figure 1** demonstrate that in GBM (at least in the majority of the patients) Endocan is expressed in endothelial, but not in GBM cells. In the case of the cancer cells, we think that it is unlikely that extracellular (endothelial cell-derived) Endocan can pass through the plasma membrane of GBM cells, enter the nucleus, and affect β -Catenin signaling. In the case of the endothelial cells, indeed, it is possible that a fraction of Endocan is not exported from the cell, remains in the cytoplasm, and can be transported into the nucleus to activate β -Catenin signaling. Importantly, there is data indicating that β -Catenin signaling in endothelial cells promotes angiogenesis in postnatal brain (Arterioscler Thromb Vasc Biol. 2019, 39(11):2273-2288). This observation may partly explain decreased neovascularization that we observed in ESM1 KO mice. We included this information in the discussion section (page 13). More detailed investigation of the mechanisms underlying differences between endothelial cells from normal brain (that express relatively low level of ESM1) and GBM-associated endothelial cells (that express high level of ESM1) will be an important area for future study.

5. What is the rationale for comparing only the genes upregulated by recombinant Endocan/PDGF-BB (Fig.4E)? Surely these treatments result in down-regulated genes as well. A more comprehensive analysis of total differentially expressed gene sets in these treatments would be warranted.

Reply: We apologize that we did not show the downregulated gene sets. GSEA analysis of both downregulated and upregulated genes is now shown in the main figures (**Reply Figure 4;** manuscript Figure 4F) as suggested by the reviewer.

Reply Figure 4: Heatmap showing normalized enrichment scores (NES) calculated from RNAseq data of 1079 cells treated with 10ng/ml rEndocan or rPDGF-BB for 48 hours, n=3 biological replicates.

6. It's unclear from the manuscript whether the ATAC-seq was performed on sphere cultures of WTD/KOD 7080 cells or from freshly isolated tumors without further culture. The results state that "...we performed ATAC-seq with Esm1 WTD and KOD cells" while the Fig. 5 legend states "ATAC-seq analysis of tumors..."; the methods section invokes Ref.39 in which ATAC-seq was performed on gliomaspheres. Also, the ATAC-seq analysis section is incomplete. Please specify which version of the mouse genome the data is aligned against and how differential chromatin accessibility was determined. The ATAC-seq methods also state that "UCSC Genome Browser was used to determine whether open regions displayed H3K27ac and conserved TF binding sites." How was this analysis used in this manuscript?

Reply: We apologize for our failures to adequately describe the experiments. ATAC-seq was performed on freshly isolated tumors without further culture. We corrected the results section accordingly. We also added the following information into the materials and methods section to better describe our data analysis strategy: "The quality assessment of the initial sequence data was conducted utilizing FastQC v0.11.9 in conjunction with multiQC v1.10.1. Paired-end reads were trimmed with Trimmomatic v0.39, employing parameters (SLIDINGWINDOW:4:15, MINLEN:35). Following trimming, the reads were aligned to the mouse genome assembly GRCm38.p6 using Bowtie2 v2.3.5.1, with the specified parameters '--very-sensitive -k 5'. Subsequently, unmapped reads and those corresponding to mitochondrial DNA were excluded from further analysis. PCR duplicates were removed utilizing sambamba v0.6.8. The resultant bam files were sorted and indexed using samtools v1.1. ATAC-Seq peak calling was performed using Genrich v0.5 under the ATAC-Seq mode. Annotation of the identified peaks were performed using the R package ChIPseeker. If a peak overlaps with the promoter region (-1000bp, +1000bp) of any transcription start site (TSS), it is annotated as a promoter peak of the gene. If a peak does not overlap with any known gene promoters or other gene-related elements it is annotated as a "Distal" peak. Peak ranges for motif analysis were written out as fasta using bedtools getfasta v2.27.1. Motif enrichment analysis was carried out with meme-chip using the Jaspas 2024 database. Coverage tracks in bigwig format were generated using the deepTools package v3.5.1."

7. The authors identify 38 genes containing differentially accessible chromatin between WTD and KOD samples. Presumably these genes harbor differentially accessible chromatin regions within the gene or at some set distance (not specified) from the TSS. How many differentially accessible peaks were identified overall? There may be a significant number of differentially accessible intergenic sites contained in potential long-range enhancers.

Reply: As indicated in Supplementary Table 3 the analysis identified a total of 529 differential peaks. Among these, 143 were associated with promoter regions, while 181 peaks, indeed originated from intergenic regions. We agree with the reviewer's comments that some of these distal peaks may represent long-range enhancers. However, it is important to mention that enhancers (unlike promoters and protein coding sequences) are poorly conserved between humans and mice and are also highly tissue specific (Cell. 2015, 160(3):554-66; Genome Biol. 2021, 22(1):62). We were unable to find any datasets containing information about enhancers in mouse brain cancer cells and therefore we could not reliably determine if the intergenic peaks identified in our ATACseq experiment correspond to the long-range enhancers.

8. Fig. 5A legend “distal” should be defined. Likewise, what constitutes promoter proximal should also be defined.

Reply: We again apologize for our lack of adequate description. If a peak overlaps with the promoter region (-1000bp, +1000bp) of any transcription start site (TSS), it is annotated as a promoter peak of the gene. "Distal intergenic" refers to regions that do not overlap with any known gene promoters or other gene-related elements. We added this information into material and methods section.

9. While it's very impressive that ATAC-seq identifies open chromatin structure of MYC promoters (and, also potential enhancers/introns) in tumors developed in ESM1 WT but not ESM1 KO animals, it is surprising the genomic regions with E-box sequences (MYC sites) is not enriched (at least not reported here). How do the authors think that ESM1 signaling only creates a very small amount of chromatin alteration? Did the authors perform *de novo* motif analysis at differentially accessible sites? If other TF binding sequences are also enriched, the authors should report those as well. Do the authors find any significant changes in H3K27ac and H3K4me3 in cells used for ATAC-seq?

Reply: We appreciate this comment. We have now performed *de novo* motif analysis of the detected ATAC-seq peaks as described in the Methods section. Although we didn't identify MYC binding motif in our dataset, the top hits in this analysis were the motifs for PATZ1 (E-value = 9.6×10^{-83}) and Irf1 (E-value = 1.2×10^{-80}) - transcription factors whose expression is regulated by MYC (Mol Cell Biol. 2015, 35(10): 1741–1753; Oncogene. 2022, 41(48):5160-5175) (**Reply Figure 5**; manuscript Figure S5C). Although it would be interesting to also compare H3K27ac and H3K4me3 profiles in tumors grown in *Esm1* WT and KO animals, we believe that this experiment goes beyond the scope of the present study.

Reply Figure 5: DNA motifs over-represented in the ATACseq dataset generated from tumors derived from *Esm1* WT and KO mice (n=2 mice per group). Transcription factors that binds to the corresponding motif and E values are indicated.

10. The spatial identity and tumor heterogeneity (as to why animal survival was significantly reduced only when both KOD or WTD cells are injected together but not alone) claims (good suggestions though) are speculative not supported by data provided and thus need to be modified.

Reply: We agree that our wording may have been over-speculative. We rewrote this part of the results: “However, when a mixture of KOD and WTD cells was co-injected into the brain, animal survival was substantially reduced compared to the mice injected with either KOD or WTD cells alone. One possible explanation of this result could be that mixing of KOD and WTD cells together enhances cancer cell heterogeneity within the developing tumors, which has been shown to promote GBM malignancy by various molecular mechanisms^{1,2}. However, additional experiments are needed to test this hypothesis.”

11. Fig S4c can be moved to main figure. Also pertaining to Fig S4c, it would appear from the GSEA results, that rEndocan activates MYC targets better than rPDGF-BB. Did rPDGF-BB induce MYC mRNA in 413 cells to a similar level as rEndocan? The western blot in Fig. 6c suggests both rEndocan and rPDGF-BB can induce MYC protein to similar levels although this blot was done in 1079 cells while the RNA-seq was performed in 413 cells.

Reply: We moved Fig S4c to the main figures. We apologize for the inconsistency in the cell lines, some of which was due to authors moving laboratories. We have performed western blot and qRT-PCR analysis for both 413 and 1079 cells treated with rPDGFbb and rEndocan. The results of this experiment demonstrated that in both cell lines rEndocan upregulates MYC at mRNA and at protein level. Interestingly, in 413 gliomaspheres rEndocan has a stronger effect than rPDGFbb, while in 1079 cell both ligands have a similar level of effect (**Reply Figure 6A-B**; manuscript Figure 5F and S5D). Based on this data, we can conclude that in general Endocan and PDGFbb have similar effects on GBM cells, however, the exact magnitude of the changes induced by each protein may be dependent on the properties of each cell line and we hesitate to overinterpret observed differences.

Reply Figure 6: A, qRT-PCR analysis of MYC expression in control and rEndocan (10ng/ml) or rPDGFbb (10ng/ml) treated cells human GBM cells (1079 and 413 cell lines). ***P<0.001, ****P<0.0001, n=3 independent experiments. B, WB analysis of samples as in “A”.

12. What happens to GBM cells when Dox inducible ESM1 is expressed? Do they become independent of endothelial cell secreted ESM1?

Reply: We thank reviewer for this interesting question. We believe that performing this experiment is beyond the scope of the current study

13. Discussion is long and can be shortened.

Reply: We thank the reviewer for this suggestion. Wherever reasonable, we have edited the discussion. However, our findings are quite numerous and the interpretations are complex, warranting a somewhat more in-depth discussion than in some other manuscripts.

14. Fig. 4: It is interesting to note that while both ESM1 and PDGF-BB activates PI3K, p42 is only activated under PDGF stimulation. What’s the biological significance considering pathways controlled by p42/44.

Reply: We added the following information into the results section: “Comparing the effects of Endocan and PDGFbb on GBM cells we noticed substantially higher levels of ERK1/2 (p44/p42 MAPK) phosphorylation upon incubation with PDGFbb. ERK1/2 activation is an integral part of the canonical PDGFRA RTK signal transduction pathway (PMID: 24359404) that was shown to promote mitotic progression and proliferation (PMID: 19331140, PMID: 16820883). Weaker effect of Endocan on ERK1/2 may be associated with the different kinetics of its interaction with PDGFRA (Sci Rep. 2017, 7(1):16439). However, further studies are needed to clarify the significance of the potentially differential activation of the ERK pathway by Endocan and PDGFbb.”

15. The actin loading control for Fig.5f is problematic. I don't think this represents a bona fide difference between WTD and KOD cells though since actin in Fig. 5e looks fine.

Reply: We have replicated this experiment three times with consistent results. We replaced the previous western with another replicate that has a better loading control (**Reply Figure 7**; manuscript Figure 5E).

Reply Figure 7: Western blot analysis of WTD and KOD cells treated or untreated with rEndocan (10ng/ml) for 3 days.

16. Figs 5G,5I, 6B, 6C, 6D, etc. legends should indicate how much rEndocan, rPDGF-BB, ponatinib were used.

Reply: We have corrected the manuscript accordingly.

17. Fig 6a: A better/cleaner blot for MYC is preferred. Are these cells treated with rESM1?

Reply: We repeated this experiment multiple times and replaced the figure with the cleaner western blots (**Reply Figure 8**; manuscript Figure 6A). In this case, cells were not treated with rESM1.

Reply Figure 8: Western blot analysis of 1079 GBM cells after lentiviral mediated knockdown of PDGFRA with 4 different shRNAs against PDGFRA or Non Target shRNA (NT) as a control.

18. How does ESM1 signaling interface with IBSP (vascular enriched integrin binding sialo protein) function in GBM regarding cell proliferation, migration, and mesenchymal transition the authors reported in Cell Rep, 2022 paper. It appears that these two factors may regulate overlapping cellular phenotype.

Reply: We thank the reviewer for this interesting question. As we mentioned at the beginning of the discussion section, VE and GBM cells form highly complex bidirectional multilayered interaction network and ESM1 together with VEGFA, IBSP and many other proteins collectively shape the phenotype of both GBM and VE cells. To better understand the relationship between ESM1 and IBSP we analyzed a recently published spatial transcriptomic dataset of GBM tumor tissues (Neurooncol Adv. 2023, 5(1):vdad142). Interestingly, we observed that while in some patients both genes are expressed in the same areas, in others they show little or no overlap (**Reply Figure 9**). Therefore, we can hypothesize that the integrated effect of VE cells on GBM cells will depend on the exact concentrations of ESM1, IBSP as well as other secreted proteins in the given region of the tumor. We have elected to not include these findings in the revised manuscript.

Reply Figure 9: Expression of ESM1, VWF and IBSP in GBM samples from previously published Visium dataset (Neurooncol Adv. 2023, 5(1):vdad142) (n = 5 different patients).

19. Does IDH (WT/mutant) phenotype influences ESM1 function in GBM? The authors should inform the IDH status of the patients samples used and include relevant sentences in discussion.

Reply: All GBM cells used in this study were IDH WT. We have now indicated this in the Methods and also describe in the Discussion that one of the limitations of our study is that we didn't investigate ESM1 signaling in IDH mutant glioma cells.

20. Considering that radiation promotes vascular endothelial and pericyte-like phenotype in GBM, does ESM1/endocan treatment alters these cellular phenotypes?

Reply: We appreciate this interesting question from the reviewer and the fact that the reviewer is aware of our prior work. We assessed the expression levels of endothelial marker CD31 and pericyte markers Desmin and ACTA2 in GBM cells from two different patients irradiated in the presence or absence of rEndocan (**Reply Figure 10**). Consistent with our recently published results (Nat Commun. 2022; 13: 6202) we detected substantial upregulations of all markers after irradiation, however rEndocan didn't show any reproducible effect on radiation induced vascular transdifferentiation under these experimental conditions. We have not included these data in the revised manuscript.

Reply Figure 10: qRT-PCR analysis of CD31, ACTA2 and Desmin expression in non-irradiated or irradiated (8Gy) gliomaspheres (1079 and 611) cultivated in the presence or absence of rEndocan (10ng/ml). n=3 independent experiments.

Reviewer #3 (Remarks to the Author):

The manuscript is very mature and complete; the data is presented very well, the statistics is sound.

I have only one single point: Fig. S7g: is there a quantification of this dataset? (If yes, please add; if not, the survival data in Fig. 7 and these exemplary images are sufficient).

Reply: We greatly appreciate the positive evaluation of our study by the reviewer. Unfortunately, for Fig. S7g we sacrificed only one representative animal per group (all at the same time point after the injection of the GBM cells) to illustrate the differences in the tumor size. The remaining animals were left alive longer to obtain the Kaplan-Meier survival curve and were sacrificed at different time points of the experiment when neurological symptoms developed. Therefore, considering that for Fig. S7g we have just one representative animal per group we believe that the quantification will not be helpful. We added this information into the figure legend.

Reviewer #4 (Remarks to the Author):

Taken together, the authors employed multidisciplinary approaches to investigate the crosstalk between tumor vasculatures and GBM cells. These findings are interesting and clinically significant. Their data also provide novel mechanistic insights into GBM progression and invasion. I have several comments that may help authors to improve their findings:

1. It is known that Endocan is a specific marker for endothelial tip cells and its expression is unregulated in a number of tumors other than GBM. It is also known that VEGF, FGF and other cytokines upregulate Endocan production in tumor vascular endothelial cells, which is correlated with tumor hypervascularization. I assume that these known functions also exist in GBM as reported in other tumor types. On the basis of these published known effects, the authors should move the results related to these data to the Supplemental Information. In this way, their rest of findings would highlight the novelty.

Reply: Based on the issues raised by Reviewers 1 and 2, we believe that it is important to demonstrate endothelial expression of Endocan in GBM with a high degree of certainty and therefore have elected to keep this information in the main figures.

2. The key surprising findings is that Endocan bindings to PDGFRA and trigger the subsequent signaling cascade. These are totally unexpected findings. The authors should emphasize these key findings in the abstract, significance and elsewhere throughout the manuscript. As an expert in the field, this reviewer have a number of questions, including: a) The PDGF ligands are made as dimers through disulfide bonds. All members within the PDGF family and VEGF family belong to this type of growth factor ligands. Endocan as a heparan sulfate proteoglycan (HSPG) share absolutely no structural similarities with PDGFs. How would they explain that Endocan could even bind to and trigger dimerization of PDGFRs. I do not know any mechanisms to explain this. The only possibility of binding to PDGFRA is through the heparin binding property

of endocan. PDGF-BB and other ligands also possess heparin-binding properties, which allow them to interact with the low affinity receptors (HSPGs). Would it be possible for Endocan to interact with PDGFRA through heparin binding? What happens if the authors perform an experiment by saturating PDGFRA heparin binding, followed by endocan binding? 3) If heparin binding is involved as a low affinity, it may help PDGFRA interact with its specific ligands and thus enhancing downstream signaling.

Reply: We greatly appreciate this insightful comment from the reviewer which has helped us to substantially improve our manuscript. To address this question, we performed additional experiments and revised the manuscript accordingly.

1) We performed a surface plasmon resonance (SPR) study to determine the K_D with which recombinant His₆-Endocan binds to recombinant Fc-PDGFR α fragment (**Reply Figure 11A**; manuscript Figure 4B). We find the $K_D = 345 \pm 25$ nM. This value is of the same magnitude as the K_D of PDGFRA binding to PDGFa-a (600nM), to PDGFb-b (420 nM) and to PDGFa-b (840 nM) which were previously determined using a similar SPR assay (Sci Rep. 2017, 7(1):16439). Importantly, in our experiment, we did not detect the binding of His₆-Endocan to control recombinant Fc fragment.

Reply Figure 11: BIAcore signal response kinetics studies of interactions between His₆-Endocan and Fc-PDGFR α (red and yellow; two biological replicates) and between His₆-Endocan and Fc (grey), (B) His₆-Endocan and Fc-PDGFR α in a presence or absence of the indicated concentrations of Heparin. (C) WB analysis of 413 cells incubated with rEndocan (10ng/ml) in the presence or absence of the indicated concentrations of Heparin.

2) To assess the potential role of heparin in Endocan-PDGFR α interaction, we first used SPR to compare the interaction of His₆-Endocan with the Fc-PDGFR α fragment in the presence or absence of heparin (**Reply Figure 11B**). We did not detect any effect of heparin on the binding of these proteins. Next, we compared the effect of Endocan on GBM cells in the presence or absence of heparin (**Reply Figure 11C**) and found that heparin has no effect on Endocan-mediated PDGFR α activation. Taken together, these results make it unlikely that the interaction of Endocan with PDGFR α is due to its heparin-binding capacity. We have not included these data in the already lengthy revised manuscript.

3) To further understand the potential relationships amongst Endocan and other PDGFR α ligands, we analyzed a recently published spatial transcriptomic dataset of GBM tumor tissues (Neurooncol Adv. 2023, 5(1):vdad142). We found that ESM1 and PDGFB are expressed in the same tumor regions, while PDGFA and PDGFC seem to be mutually exclusive with PDGFB and ESM1 (**Reply Figure 12A-B**; manuscript Figure S5A-B). The potential reasons for this are unclear, but we speculate that Endocan may act together with PDGFB to maintain a high level of PDGFR α activation in the same tumor region, while other PDGFR ligands are expressed in the different GBM areas and may therefore affect other populations of GBM cells.

Reply Figure 12: **A**, Expression of ESM1, VWF, PDGFA, PDGFB and PDGFC in GBM samples from previously published Visium dataset (Neurooncol Adv. 2023, 5(1):vdad142) (n = 5 different patients). **B**, Quantification of ESM1 expression colocalization with the expression of VWF, PDGFA, PDGFB and PDGFC in samples as in “B”. Data points related to different patients are indicated in different colors. Wilcoxon signed rank-sum test, *P<0.05.

In their experimental system, the PDGF ligands always present because the serum contains high concentration of PDGFs, including PDGF-AA, PDGF-BB and PDGF-AB related by platelets during coagulation. They should completely remove PDGFs in order to make this conclusion.

Reply: As indicated in the materials in methods all experiments with GBM cells were performed in serum free medium using standard gliomasphere cultivation protocol with PDGF ligands only added as indicated.

4) They should provide K_D of Endocan-PDGFRa binding;

Reply: As indicated above, the K_D of Endocan-PDGFRa binding = 345 ± 25 nM.

5) The TKI drugs for the type of experiments is inappropriate because of their broad effects. Specific neutralizing antibodies against PDGFRa should be used.

Reply: This is an important point. The TK inhibitors were used to provide potential translational relevance to our finding. However, we acknowledge that these do not prove that Endocan produces its phenotypic effects via interaction with PDGFRa. To more specifically support our hypothesis, in the original manuscript (Figure 6b), we demonstrated that knockdown of PDGFRa inhibited the proliferative effects of Endocan on GBM cells. As per the reviewer's comments and to provide additional robustness to our data, we have now demonstrated that preincubation of patient-derived gliomaspheres with olaratumab – a widely used neutralizing antibody against PDGFRa, prevents Endocan and PDGFbb-induced phosphorylation of PDGFRa (**Reply Figure 13A**; manuscript Figure S6C) and blocks Endocan mediated upregulation of Myc expression (**Reply Figure 13B**; manuscript Figure S6D).

Reply Figure 13: **A**, WB analysis of 1079 cells incubated with rEndocan or rPDGFbb for the indicated period of time in the presence or absence of olaratumab (9 nM). **B**, qRT-PCR analysis of MYC expression in 1079 cells cultivated for 3 days with rEndocan in the presence or absence of olaratumab (9 nM). n=3 independent experiments.

The reason why I ask for these additional experiments to strengthen their conclusions is that their claim of Endocan in binding and triggering PDGFRA is very provocative in this research field. The authors need to provide convincing and robust experimental data to support this conclusion. I hope this would be helpful for them to improve the quality of their work.

3. The radiation therapy part should be omitted from this manuscript because of the flow and mechanistic link is too vague.

Reply: We understand that the radiation experiments do not provide additional mechanistic insight. However, we strongly believe that this part of the manuscript is very important due to its potential translational impact. First, the strongest effect of Endocan *in vivo* was associated with altered radiosensitivity of murine GBM-like tumors (**Reply Figure 14A**; manuscript Figure 7A). Second, radiation more than doubles Endocan secretion by VE cells (**Reply Figure 14B**; manuscript Figure 7F), indicating that in real tumors the strongest (and possibly the most oncogenic) effect of this protein may be observed during or after radiation therapy.

Reply Figure 14: **A**, ELISA assay comparing levels of secreted Endocan in conditioned media collected from HBEC-5i cells, TEC14, and TEC15 cells following irradiation with 8 Gy at 48-hour post radiation time-point. n=2 independent experiments. *P=0.018, **P=0.0332; ***P=0.0333. **B**, Kaplan Meier survival analysis of Esm1 WT or Esm1 KO mice intracranially injected with 7080 cells and subsequently irradiated with 3 doses of 2.5 Gy 14 days after injection. n=5 mice per group. P-value was determined by log-rank test, ns – non significant; *P=0.00875; **P=0.00443; ***P=0.00206.

4. The authors should consider to cite the following publications:

PMID: 27608497

PMID: 18392794

PMID: 23773831

I hope that my comments are helpful for improving the quality of the authors' otherwise very important work. I look forward to reading a revised version.

Reply: We thank the reviewer for directing us to these important papers. In fact, findings described there are consistent with the new results that we obtained by analyzing spatial transcriptomic dataset (**Reply Figure 12A-B**). Using this approach, we demonstrated that ESM1 and PDGFB are expressed in the same tumor regions and therefore both proteins may affect the same population of GBM cells, promoting consistent phenotypic changes. We added the following text into the discussion section: “Surprisingly, we found that ESM1 and PDGFB are expressed in the same tumor regions, while PDGFA and PDGFC seem to be mutually exclusive with PDGFB and ESM1. Analysis of the literature shows that Endocan mimics many of the functional capabilities of PDGFbb as prior studies have shown PDGFbb as a key regulator of GSC self-renewal similar to Endocan’s pro-proliferative and tumorigenic properties^{3,4}. In regard to vascular remodeling, PDGFbb, much like Endocan, is highly expressed in endothelial tip cells during neoangiogenesis, regulates vessel formation⁵ and permeability thereby contributing to leaky tumor vasculature and enhanced tumor growth⁶⁻⁹. Findings listed above highlight a potential synergistic relationship between PDGFR ligands in GBM. It will be of great interest to determine differences in the effects of various PDGFR ligands on glioblastoma cells and to identify cells secreting these ligands within the tumor.”

References:

1. Ishihara, H. *et al.* Endothelial Cell Barrier Impairment Induced by Glioblastomas and Transforming Growth Factor β 2 Involves Matrix Metalloproteinases and Tight Junction Proteins. *J. Neuropathol. Exp. Neurol.* **67**, 435–448 (2008).
2. Nefitel, C. *et al.* An Integrative Model of Cellular States, Plasticity, and Genetics for Glioblastoma. *Cell* **178**, 835-849.e21 (2019).
3. Jiang, Y., Boije, M., Westermark, B. & Uhrbom, L. PDGF-B Can sustain self-renewal and tumorigenicity of experimental glioma-derived cancer-initiating cells by preventing oligodendrocyte differentiation. *Neoplasia N. Y. N* **13**, 492–503 (2011).
4. Cao, Y. Multifarious functions of PDGFs and PDGFRs in tumor growth and metastasis. *Trends Mol. Med.* **19**, 460–473 (2013).
5. Hosaka, K. *et al.* Pericyte-fibroblast transition promotes tumor growth and metastasis. *Proc. Natl. Acad. Sci. U. S. A.* **113**, E5618-5627 (2016).
6. Cao, Y., Cao, R. & Hedlund, E.-M. R Regulation of tumor angiogenesis and metastasis by FGF and PDGF signaling pathways. *J. Mol. Med. Berl. Ger.* **86**, 785–789 (2008).
7. Guo, P. *et al.* Platelet-derived growth factor-B enhances glioma angiogenesis by stimulating vascular endothelial growth factor expression in tumor endothelia and by promoting pericyte recruitment. *Am. J. Pathol.* **162**, 1083–1093 (2003).
8. Hellström, M., Kalén, M., Lindahl, P., Abramsson, A. & Betsholtz, C. Role of PDGF-B and PDGFR-beta in recruitment of vascular smooth muscle cells and pericytes during embryonic blood vessel formation in the mouse. *Dev. Camb. Engl.* **126**, 3047–3055 (1999).
9. Gerhardt, H. *et al.* VEGF guides angiogenic sprouting utilizing endothelial tip cell filopodia. *J. Cell Biol.* **161**, 1163–1177 (2003).

Point-by-point Response to Reviewers' Comments

Reviewer #2 (Remarks to the Author):

Comments on revised manuscript:

We feel that the authors have sincerely responded to the critiques of all reviewers, resulting in revised text and inclusion of new results and figures. The manuscript should be acceptable once the following changes are implemented.

1. Fig. 4E requires modification and explanation. The immunoblot in Fig.4E purports to show 10 total samples but only the top three blots (p-PDGFR α , PDGFR α , bActin) contain 10 lanes. The next 5 blots contain only 9 lanes.

Reply: We sincerely apologize for this error. We corrected the corresponding figure and provided uncropped scans of all blots.

2. Response of authors to reviewers 1 and 2: "Consistent with our recently published results (Nat Commun. 2022; 13: 6202) we detected substantial upregulations of all markers after irradiation, however rEndocan didn't show any reproducible effect on radiation induced vascular transdifferentiation under these experimental conditions. We have not included these data in the revised manuscript.

Comment: The authors should include these results as supplemental data in the manuscript as irradiation studies are included in the manuscript (Fig 7) and these results are relevant in that context.

Reply: According to the reviewer request we included these data as **Supplementary Figure 7f**.

3. Response of authors to reviewer 3: "Taken together, these results make it unlikely that the interaction of Endocan with PDGFR α is due to its heparin-binding capacity. We have not included these data in the already lengthy revised manuscript."

Comment: We believe Reply figure 11B and C need to be included in supplemental data for new Fig 4B as a key control that addresses reviewer 3's questions regarding binding specificity.

Reply: According to the reviewer's request we included these data as **Supplementary Figures 4d-e**.

4. Discussion continues to remain extremely long.

5. It would have been easier if the authors highlighted the text in the manuscript where major changes were made.

Reply: We shortened the discussion and underlined all changes that were introduced into the manuscript during this revision. We continue to feel, however, that the extent and novelty of our findings warrant a more thorough discussion than might be typical in Nature Communications or other similar journal.

6. Fig 8 can be included as Fig 7G.

Reply: We believe that Figure 7 already contains a substantial number of panels and subpanels, occupying nearly the whole page. We would prefer not to decrease the size of the individual panels which can make our results more difficult for the reader to grasp. Therefore, we elected to include the final schema as a separate figure rather than as an additional part of Figure 7.

Reviewer #5 (Remarks to the Author):

Bastola et al. "Endothelial-secreted Endocan acts as a PDGFR alpha ligand..." (NCOMMS-23-44699A)

I think this paper reports an interesting finding and that the authors overall have dealt satisfactory with the criticism from the reviewers. However, the point raised by reviewer 4 regarding the affinity by which Endocan binds to PDGFR alpha and whether it induces receptor dimerization have not been satisfactory addressed.

Reply: We address the specific points, including the comment on binding affinity below but also want to take the opportunity to summarize our views here. In this manuscript, including the additional experiments described below and prompted by this reviewer, we used multiple independent approaches to compare the effects of Endocan and PDGF-BB: 1) WB analysis of PDGFRA phosphorylation 2) SPR assay, 3) PLA, 4) binding competition experiment, 5) RNAseq, 6) RT-qPCR, 7) cell growth assay, 8) inhibition of PDGFRA with small molecules (Ponatinib or Nintedanib) or with neutralizing antibodies (Olaratumab). Furthermore, we examined the effects of Endocan following knockdown of PDGFRA. These experiments show that Endocan and PDGF-BB have similar qualitative effects in the biologically relevant concentration ranges of both ligands and clearly demonstrate that abrogation of PDGFRA signaling prevents a majority of the biological effects of Endocan. Thus, despite the lack of direct evidence that Endocan induces PDGFRA homodimerization and the presence of some quantitative differences between Endocan and PDGFBB effects, the most parsimonious explanation for our data is that the majority of effects of Endocan described in this manuscript are mediated via direct interaction with PDGFRA.

1. The authors have used SPR to determine the K_d of Endocan binding to PDGFR alpha. They have come up with a value of 345 nM, which they claim is in the same range as previously has been reported by Mamer et al. (2017) Sci.Rep. (420 nM for the binding of PDGF-BB to PDGFR alpha; 660 nM for the binding of PDGF-AA to PDGFR alpha), who also used SPR for the analysis. However, these values determined by SPR are very different from those determined by conventional Scatchard analyses. Thus, values of 0.2 nM for PDGF-AA and 0.5 nM for PDGF-BB binding to PDGFR alpha on living cells, were reported by Claesson-Welsh et al. (1989) PNAS 86, 4917-4921. It should be noted that the values determined by Scatchard analysis corresponds nicely to the half-maximum effect by PDGF-AA and PDGF-BB on receptor activation measured as autophosphorylation and cell proliferation. Thus, the SPR data are artifactual. A possible reason could be that on living cells with a possibility for movement of the receptors in the plasma membrane, a dimeric PDGF molecule binds to two receptors, whereas in the SPR analysis the receptors are immobilized, thus possibly preventing dimeric interactions and as a consequence giving values indicating about a thousand-fold lower affinity. The authors need to perform a regular Scatchard analyses on living cells, comparing the binding of Endocan and PDGF-BB.

Reply: In the manuscript mentioned by the reviewer, the authors evaluated the binding of a constant amount of radioactively labeled PDGF-BB to the cells exogenously transduced with PDGFRA in the presence of various concentrations of unlabeled PDGF-BB. We have already performed a similar assay (**Figure 4C and D**). In our experiment, we evaluated the binding of a constant amount of fluorescently-labeled Endocan to patient-derived glioma cells in the presence of various concentrations of PDGF-BB. We also did the opposite assay where we evaluated the binding of a constant amount of fluorescently labeled PDGF-BB in the presence of various concentrations of Endocan.

Although our experiment does not allow for the precise calculation the Endocan-PDGFR α binding constant, it demonstrates that both ligands primarily bind to the same receptor and that Endocan and PDGFF-BB have similar affinities for PDGFR α . These findings have the advantage that we are investigating binding to endogenous receptor(s). To make our results easier to comprehend we replotted the data in a form similar to the Scatchard assay (**Reply Figure 1**). The graphs indicate that Endocan has similar or even higher affinity for the cellular receptor than does PDGFF-BB. Interestingly, a concentration of 10 μ g/ml rEndocan was able to completely displace rPDGF-BB from GBM cells, while rPDGF-BB at the corresponding concentration was able to displace only half of the rEndocan, suggesting that Endocan can theoretically interact with other receptors in addition to PDGFR α .

Reply Figure 1: Competition binding assay of Endocan and PDGF-BB. **A**, 1079 cells were incubated with Endocan-Alexa488 (10 μ g/ml) in the presence of various concentrations of PDGF-BB-Alexa647 (0.1-10 μ g/ml). **B**, 1079 cells were incubated with PDGF-BB-Alexa647 (10 μ g/ml) in the presence of various concentrations of Endocan-Alexa488 (0.1-10 μ g/ml). After incubation and extensive washing binding of fluorescently labeled proteins to the cell surface was analyzed by FACS.

2. The authors show a PLA-based analysis of receptor dimerization in response to Endocan binding, with PDGF-BB as control. However, it is not clear exactly how this analysis was done and what the values on the y-axis mean (“Dimerization intensity”). The authors need to perform a conventional cross-linking analysis, comparing dimerization in response to binding of Endocan and PDGF-BB.

Reply: We apologize for not providing sufficient details for this experiment. Patient-derived 1079 glioblastoma cells were plated in Lab-Tek II chamber wells pre-coated with laminin. The next day, cells were treated with rEndocan (10ng/ml) or rPDGF-BB (10ng/ml) for 90 minutes, washed 3 times with ice cold PBS and fixed with 4% PFA in PBS for 15 min at room temperature. Next, cells were washed 2 times with PBS and permeabilized with 0.2% Triton-X100 in PBS for 15 minutes. Subsequent procedures were performed using “Duolink In Situ Orange Starter Kit” (Duolink) according to the manufacturer’s protocol. Dual recognition primary antibody pair set (anti-PDGFR α and anti-p-PDGFR α) validated for PLA analysis was used (#DP0013, Abnova). We included this information into the materials and method section. We also changed the name of y-axis to “Number of PLA foci per cell” (Mol Ther Nucleic Acids. 2019, 14:52-66). We have provided these details in the amended Methods section.

According to the reviewer’s suggestion, we also attempted to perform the cross-linking analysis using the previously described protocol (Cancer Res, 2010, 70(10); 4195–203). Unfortunately, this assay appeared to have very low sensitivity as even in our positive control (GBM cells treated with 20ng/ml rPDGF-BB) we could barely see a band that may possibly correspond to the PDGFR α dimer (**Reply Figure 2**). Analysis of the literature revealed that after crosslinking, a substantial amount of PDGFR dimer was only detected in the cells that overexpress exogenous PDGFR (Oncogene. 2014, 33(20):2568-76; Proc Natl Acad Sci U S A. 2008, 105(22):7681-6), while in the cells with endogenous PDGFR, the crosslinking assay showed relatively minor degree of PDGFR dimerization (Cancer Res, 2010, 70(10); 4195–203; Exp Cell Res. 2019, 380(1):69-79). Based on these data and our results we can speculate that in our experimental model the cross-linking analysis may not be sensitive enough to reliably detect PDGFR α activation.

However, despite these data, we believe that our other results, which were obtained using multiple independent approaches (WB analysis, SPR assay, PLA, binding competition experiment, RNAseq, RT-qPCR, cell growth assay, inhibition of PDGFRA with small molecules (Ponatinib or Nintedanib) or with neutralizing antibodies (Olaratumab)) have already demonstrated that Endocan can activate the PDGFRA pathway in GBM cells.

Reply Figure 2: 1079 and 413 cells were incubated for 30 minutes with 20ng/ml of rEndocan or rPDGF-BB and thereafter the surface proteins were cross-linked on ice, separated on PAAG and immunoblotted for PDGFRA. Experiment was performed in n = 3 biological replicates.

3. There is also room for improvement of the immunoblots shown. In Fig. 4e, the lanes are poorly aligned, and there are several P-PDGFR bands (also the case for other figures); markers of molecular mass should be inserted so that one can compare with the blot for PDGFRA (this goes for all immunoblots). Fig. S4b suggests that PDGF-BB causes dramatically more auto-phosphorylation than Endocan. However, quantification is difficult since the authors show only a kinetic analysis. The authors need to perform dose-response analyses, comparing the auto-phosphorylation of PDGFR alpha in response to different concentrations of Endocan and PDGF-BB.

Reply: We sincerely apologize for mislabeling this immunoblot and thank the reviewer for pointing it out. We corrected the corresponding figure and provided uncropped scans of all blots.

According to the reviewer's suggestion, we performed dose-response analyses by treating 1079 patient-derived GBM cells with different concentrations of rEndocan and rPDGF-BB for 30 minutes (**Reply Figure 3**). Results of this experiment demonstrated that both rPDGF-BB and rEndocan induced PDGFRA phosphorylation in brain tumor cells at sub-nanomolar concentrations, which are consistent with the Kd value previously identified for PDGF-BB*PDGFRA interaction using Scatchard analyses (Proc Natl Acad Sci U S A. 1989, 86(13):4917-21) mentioned by the reviewer. While these data confirm that Endocan can act on the PDGFRA at concentrations similar to that of PDGF-BB, the magnitude of the biological effect may be different. We added these results as **Figure 4e** and **Supplementary Figure 4b** into the manuscript.

Reply Figure 3: Western Blot (left panel) and quantification of the results (right panel) obtained with 1079 cells incubated with different concentrations of rEndocan or rPDGF-BB for 30 minutes and analyzed for PDGFRA phosphorylation and bActin level. Experiment was performed in n = 2 biological replicates.

Reviewer #5 (Remarks to the Author):

In their response to my criticism, the authors argue that they have shown by many assays that Endocan activates PDGFR-alpha. I do not dispute this, and I do think their observations are interesting and should be published. However, the quantifications of the binding affinities are not satisfactory, neither in the original version nor in the revised version.

Reply: We thank the reviewer for the overall positive evaluation of our findings and we apologize that in this study we were unable to fully quantify the binding affinity of Endocan to PDGFRa. We agree that these data would be a significant contribution to the field. However, we feel that a detailed study of Endocan-PDGFRa interaction is well beyond the scope of this already large paper. To highlight that our study misses this information we added the following study limitation paragraph into the discussion section:

“Although we used multiple methods and models to confirm that endothelial cell-derived Endocan can interact with PDGFRa and activate downstream signaling pathways in GBM cells, our work has several important limitations. First, it is not clear how Endocan that shares no structural similarity with classical PDGFRa ligands interacts with this receptor. To address this question, the structure of the PDGFRa-Endocan complex will need to be established. Second, it will be important to determine the binding constant of the native (correctly glycosylated) Endocan with PDGFRa dimers that are present on the surface of GBM cells. Third, differences in the effects of Endocan and PDGF-BB as well as the results of our competition assay (Figure 4B) suggest that in addition to PDGFRa there might be other receptors for Endocan on the surface of GBM cells which have to be determined. Finally, it is still not clear by what mechanism Endocan-induced PDGFRa activation affects the chromatin structure of the Myc promoter region.”

1. The SPR values of the interaction affinities are artifactual (for PDGF, about a thousand-fold lower than what has been measured by Scatchard analysis, and functional assays). The authors themselves state on page 9, line 14 that both PDGF-BB and Endocan induce phosphorylation of PDGFR-alpha at sub-nanomolar concentrations. So, they contradict themselves. I strongly recommend to remove the artifactual SPR data which only causes confusion.

Reply: We agree that the SPR analysis gives K_d 's that are likely to be incorrect and are confusing. Given that the analysis is done on isolated recombinant protein fragments under non-physiologic conditions, this is not entirely surprising. However, we do think the SPR analysis adds value to the manuscript, as it shows specific Endocan binding with similar affinity as previously reported for PDGFBB and that this binding is not influenced by heparin—a question raised by previous reviewer 4. Therefore, we moved SPR data to the supplementary figures 4a and 4e and in the results section indicated in all cases whether the data are related to the recombinant monomeric PDGFRa fragment, exogenously expressed cell surface PDGFRa or endogenous PDGFRa. To clarify discrepancies between binding constants obtained for the above mentioned types of PDGFRa protein we added the following information into the results section:

“Next, we used surface plasmon resonance (SPR) to evaluate the binding affinity of recombinant His6-Endocan to a recombinant Fc-PDGFRa monomeric fragment (Figure S4a) and found a K_D of 345 ± 25 nM. Although this value is orders of magnitude higher than the one that was determined using standard Scatchard assay with radioactively labeled PDGF-BB and cells overexpressing exogenous PDGFRa, it is comparable to the K_D which were previously identified using a similar SPR approach for the interaction of a recombinant PDGFRa with PDGF-AA (600nM), PDGF-BB (420 nM) and PDGF-AB (840 nM). ... we investigated whether the addition of rEndocan alters the phosphorylation levels of endogenous PDGFRa in human gliomaspheres in vitro using PDGF-BB as a positive control. Dose-response analyses demonstrated that both rPDGF-BB and rEndocan induced PDGFRa phosphorylation in brain tumor cells at low nanomolar concentrations. This result is consistent with the K_D value previously determined for PDGF-BB interaction with exogenously expressed full length PDGFRa on the cell surface. Interestingly, the magnitude of the effect of rEndocan was somewhat smaller than that of PDGF-BB (Figure 4c and Figure S4c), which may be related to differences in ligand-induced PDGFRa internalization.”

2. Regarding dimerization, I am surprised that the conventional cross-linking assay did not work. Anyhow, the “dimerization experiment” that is shown in Fig. S4a, does not record dimerization. In the PLA assay, an antibody against PDGFR-alpha and another against p-PDGFR-alpha were used. Thus, the assay measures auto-phosphorylation of PDGFR-alpha, not dimerization. This should be clarified. In fact, there is no experiment shown that addresses the issue of dimerization. Moreover, the experimental condition used should be specified better in this experiment and in many others.

Reply: We apologize for the miswording. We corrected the result section to indicate that PLA was used to measure auto-phosphorylation of PDGFR-alpha. We also added detailed experimental conditions that were used in the experiments with GBM cells incubated with PDGFR-bb, Endocan or EGFR into the “Cell cultures” section of the Material and methods.

3. The authors now have performed a dose-response experiment and have included the result as a novel Figure 4e. The experimental conditions are not specified, so the experiment is difficult to evaluate. Was incubation at 0 C or 37 C? I suspect that it was at 37 C, since the PDGF-BB activation of the receptor goes down at the highest concentrations of PDGF-BB. It should be noted that at 37 C, one measures the net effect of activation of auto-phosphorylation of the receptor and internalization and degradation; the latter could be quite substantial after a 30-minute incubation. Better is to perform such an experiment at 0 C, when there is no internalization and degradation. Moreover, inspection of the blot reveals that PDGF-BB induces auto-phosphorylation of the receptor at 0.15 nM, which is consistent with the expectation (assuming that it is the lower band that represents the receptor; by the way, the authors need to put in molecular mass markers in the immunoblots, in order to make it possible to evaluate them; I pointed this out in my previous report). Endocan seems to give a very faint band at 0.015 nM, but no band at 0.5 nM, and a more impressive band at 1.5 and 5 nM. It is very difficult to conclude from this experiment that Endocan

activates the receptor at sub-nanomolar concentrations. The authors need to show a more conclusive experiment. The fact that different maximum levels are found for stimulation with PDGF-BB and Endocan is interesting, and may be related to differences in ligand-induced internalization, something that could easily be investigated.

Reply: We adjusted our wording and used of the word “low nanomolar” instead of “sub-nanomolar”. However, there may be a misunderstanding. It appears that the reviewer is referring to only one band in the blots. However, based on our findings with inhibitors (**Fig. S7C and S7B**) and other published studies (see <https://www.nature.com/articles/s41467-018-06949-w>, supplementary Figure 9), the phospho PDGFRA appears as multiple bands, and we used all of these bands for quantification (see dotted red line in the image below). Although the reasons for these multiple bands and the lack of appearance of the lowest molecular weight band at the 0.5 nM Endocan concentration are not entirely clear, we felt that all the bands had to be taken into account for appropriate quantitation. Based on this, and as shown in **Figure 4c** and **supplemental Figure 4c**, we see clear evidence of activation of phosphorylation at 0.5 nM. We described how quantitation was done in the supplementary figure legend and indicated molecular weight markers in all blots in the main figures, supplementary materials and in the source data file.

With respect to doing PDGFRA phosphorylation studies at 0 degrees, as assumed by the reviewer, we did do the experiments at 37 degrees and we added this information to our methods section. We chose this temperature as we were most interested in the capabilities of the ligands to activate the receptor under physiological conditions. Many studies have reported that temperature has a dramatic effect on kinase activity (FEBS J. 2021, 288(10):3148-3153) and therefore we believe that the results obtained at 37 degree will be more relevant than the similar experiment at 0°C.

Reviewer #5 (Remarks to the Author):

The authors have addressed the two first points of criticism I raised in a satisfactory manner. However, the third point has not been addressed. I reiterate for the third time that the authors must put markers of molecular masses in their immunoblots; without this information, it is not possible to evaluate the blots. In their rebuttal, the authors claim that all bands represent the receptor. The band with lowest molecular mass is absent before stimulation with PDGF-BB or Endocan and increases upon ligand stimulation, whereas the two uppermost bands are seen also without stimulation with PDGF-BB or Endocan. In the absence of any molecular mass markers, I draw the conclusion the the lower band represented the receptor (maybe I was wrong). If the uppermost bands represent the full length receptor, why is there a background activation of the receptor? If so, the lower band may be a fragment of the receptor, possibly occurring upon internalization and degradation of the receptor. If the authors had followed my advise and performed the experiment on ice, it is likely that the pattern would have been easier to interpret. The authors argue that they have chosen to perform the experiment at 37 C since this is "more relevant". Well, this depends on the question asked. At 37 C there is internalization and degradation of the receptor which complicates the interpretation. If the authors insist of performing the experiment at 37 C, they could at least have chosen a shorter time of incubation, i.e. 5-10 minutes, when internalization and degradation have barely started.

The authors have a very interesting finding to report. I am confident that they do not want to pollute their excellent paper with substandard blots. I strongly recommend the author to perform the immunoblotting experiment again.

Reply: We thank the reviewer for the overall positive evaluation of our findings. We put markers of molecular masses in immunoblots in main figures, supplementary figures and in the source data file. The presence of multiple bands, corresponding to phosphorylated PDGFRA has been described previously (Nat Commun. 2018, 9(1):4583). To further clarify this issue, we added the following text into the manuscript:

“We also observed that in some western blots phosphorylated PDGFRA appeared as multiple bands, and the lower molecular weight bands were especially noticeable when cells were incubated with rPDGF-BB, but not Endocan. These could potentially be explained by differences in PDGFRA internalization and subsequent degradation induced by rPDGF-BB and Endocan.”